# SRSF1 promotes vascular smooth muscle cell proliferation through a Δ133p53/KLF5 pathway

Ning Xie[1,2,*,†], Min Chen[1,*], Rilei Dai[1], Yan Zhang[2], Hanqing Zhao[3], Zhiming Song[4], Lufeng Zhang[4], Zhenyan Li[1], Yuanqing Feng[2], Hua Gao[5], Li Wang[1], Ting Zhang[1], Rui-Ping Xiao[2], Jianxin Wu[1] & Chun-Mei Cao[1,2,6]

Though vascular smooth muscle cell (VSMC) proliferation underlies all cardiovascular hyperplastic disorders, our understanding of the molecular mechanisms responsible for this cellular process is still incomplete. Here we report that SRSF1 (serine/arginine-rich splicing factor 1), an essential splicing factor, promotes VSMC proliferation and injury-induced neointima formation. Vascular injury *in vivo* and proliferative stimuli *in vitro* stimulate SRSF1 expression. Mice lacking SRSF1 specifically in SMCs develop less intimal thickening after wire injury. Expression of SRSF1 in rat arteries enhances neointima formation. SRSF1 overexpression increases, while SRSF1 knockdown suppresses the proliferation and migration of cultured human aortic and coronary arterial SMCs. Mechanistically, SRSF1 favours the induction of a truncated p53 isoform, Δ133p53, which has an equal proliferative effect and in turn transcriptionally activates Krüppel-like factor 5 (KLF5) via the Δ133p53-EGR1 complex, resulting in an accelerated cell-cycle progression and increased VSMC proliferation. Our study provides a potential therapeutic target for vascular hyperplastic disease.

[1] Capital Institute of Pediatrics, Beijing 100020, China. [2] Institute of Molecular Medicine, Peking University, Beijing 100871, China. [3] National Institute of Biological Sciences, Beijing 102206, China. [4] Department of Cardiology, Peking University, Third Hospital, Beijing 100191, China. [5] Center for Bioinformatics, Peking University, Beijing 100871, China. [6] Research Center on Pediatric Development and Diseases, Chinese Academy of Medical Sciences, Beijing 100730, China. * These authors contributed equally to this work. † Present address: Department of Pediatrics and Lillehei Heart Institute, University of Minnesota, Minneapolis, Minnesota 55455, USA. Correspondence and requests for materials should be addressed to C.-M.C. (email: caochunmei@pku.edu.cn).

The proliferation of vascular smooth muscle cells (VSMCs) is pivotal to intimal hyperplasia, which is a prevalent and severe pathophysiological process that contributes to arteriosclerosis, postangioplasty restenosis, vein graft stenosis and allograft vasculopathy. Under normal conditions, VSMCs maintain a non-proliferative state in the arterial tunica media. In response to injury or other stimuli, media VSMCs migrate into the intima, start to proliferate and secrete extracellular matrix, resulting in expansion of the arterial intima, that is, neointima formation[1], which can further lead to severe cardiovascular diseases, such as hypertension, ischaemic disease and subsequent myocardial infarction, stroke and congestive heart failure[1,2]. Thus, investigations into the pathophysiology and molecular mechanisms underlying VSMC proliferation and neointima formation are urgently needed.

The serine/arginine-rich (SR) proteins are essential splicing regulators required for constitutive pre-mRNA splicing and alternative splicing[3,4], while changes in the expression of SR proteins are potentially involved in splicing misregulation in various diseases[3]. Serine/arginine splicing factor 1 (SRSF1, also named ASF/SF2) is the prototypical member of the highly conserved SR protein family that functions in key aspects of mRNA metabolism, such as constitutive and alternative splicing[5], RNA polymerase II transcription[6], nuclear export of mature mRNA[7] and translation[8,9], suggesting its importance in most cell types. In addition, SRSF1 plays critical roles in the maintenance of genomic stability, cell viability and cell-cycle progression[6,10,11]. Loss of the SRSF1 gene in mouse and Caenorhabditis elegans leads to embryonic lethality[6,12,13], and deletion of SRSF1 expression in chicken DT-40 cells induces cell-cycle arrest and apoptosis[10]. Recent studies have proposed a pro-oncogenic action of SRSF1 on the basis of its upregulation in different types of human cancer and its ability to drive the oncogenic transformation of fibroblasts and epithelial cells via enhanced proliferation and compromised apoptosis when overexpressed[14–16]. Specifically, SRSF1 is highly expressed in VSMCs and is actively involved in the alternative splicing events that shape the transcriptome of proliferative VSMCs[17]. However, the function and significance of SRSF1 in vascular biology and physiology are unknown.

Here we reveal a potential role of SRSF1 in regulating VSMC proliferation and its malfunction in the pathogenesis of neointimal hyperplasia. We demonstrate that SRSF1 facilitates the migration and proliferation of VSMCs, and thus the intimal thickening, after vascular injury. SRSF1 deficiency blocks the intimal hyperplasia. SRSF1 induces a truncated p53 isoform, Δ133p53, resulting in the transcriptional activation of Krüppel-like factor 5 (KLF5) via the Δ133p53–early-growth-response gene 1 (EGR1) complex.

## Results

### SRSF1 expression increased in proliferating VSMCs.
SRSF1 is conserved in Drosophila, zebrafish, mouse, rat and human (Supplementary Fig. 1a, a molecular phylogenetic tree of SRSF1 created by MEGA6 using protein-coding nucleotide sequences[18]). Relative to the expression level in the heart, SRSF1 protein in the aorta was more abundant in rats at the age of 12 weeks (Supplementary Fig. 1b). It was stably expressed in rat aorta after birth, except for a transient increase at 2 weeks (Supplementary Fig. 1c). To define the potential role of SRSF1 in vascular biology and pathophysiology, we used an in vivo rat model of carotid artery injury and cultured human aortic SMCs (HASMCs).

First, we assessed the expression profile of SRSF1 in balloon-injured rat common carotid arteries. Mild-to-moderate intimal hyperplasia developed in the arteries at 7 days postinjury became more severe at 14 days and was sustained at 21 days. Generally

seen at low levels in the normal vascular wall, SRSF1 staining was markedly increased in the neointima in response to injury at 7 and 14 days (Fig. 1a), while the number of SRSF1-positive cells decreased at 21 days postinjury, a time point when most VSMCs are back to non-proliferating state in this carotid artery injury rat model. Dual immunofluorescence staining with specific anti-bodies against SRSF1 and VSMCs (SM-α-actin) showed that SRSF1 was predominantly localized to VSMCs in the neointima at 14 days after injury (Fig. 1b). The Srsf1 mRNA levels increased in injured arteries at 1, 4, 7, 14 and 21 days after injury (Fig. 1c). Expression of the proliferation marker proliferating-cell nuclear antigen (PCNA) was upregulated at 4, 7 and 14 days postinjury, but it decreased at 21 days (Fig. 1d). Furthermore, western blot analysis showed no higher SRSF1 in injured than in sham-operated arteries on the first day, then an increase at 4, 7 and 14 days after injury (Fig. 1d), but a decrease at 21 days compared to that at 14 days (Fig. 1d). The above results indicate that SRSF1 is upregulated and activated in VSMCs during neointima formation.

Next, using cultured HASMCs, we investigated the responses of SRSF1 to various stimuli relevant to vascular injury. Treatment of cells with angiotensin (Ang II, 200 nM) increased SRSF1 expression at the mRNA and protein levels in a time-dependent manner (Fig. 1e,f). Similarly, the expression of SRSF1 increased in response to serum (10% foetal bovine serum (FBS)) or platelet-derived growth factor BB (PDGF-BB, 10 µg l$^{-1}$) (Fig. 1e,f). Moreover, SRSF1 was also upregulated in hyperplastic arteries from human patients (Supplementary Fig. 2). These results provided the critical clue for a link between SRSF1 and VSMC hyperplasia.

### SRSF1 deficiency inhibits injury-induced neointima formation.
To assess the role of SRSF1 in VSMC function, we used SMC-specific Srsf1-knockout mice, which were derived by crossing floxed Srsf1 mice with SM22α-Cre transgenic mice (Srsf1$^{flox/flox}$ × Sm22α-Cre) (Fig. 2a). The SRSF1 expression was specifically downregulated in the aorta of Srsf1$^{-/-}$ mice (Srsf1$^{flox/flox}$; Sm22α-Cre$^{+}$) compared with wild-type (WT) littermate controls (Srsf1$^{flox/flox}$; Sm22α-Cre$^{-}$) (Fig. 2b; Supplementary Fig. 3a).

Under baseline conditions at 8–12 weeks of age, there was no difference in blood pressure or heart rate between Srsf1$^{-/-}$ mice and their WT littermates (Supplementary Fig. 3b,c), and the thickness of the media wall in uninjured sham-operated carotid arteries was comparable (Fig. 2c). SMC-specific deletion of SRSF1 overtly attenuated the wire injury-induced neointima formation at 14 and 28 days after injury (Fig. 2c,d). The average intima/media ratio in Srsf1$^{-/-}$ mice decreased to 29.56% and the neointimal area decreased to 23.05% of the value in the WT group at 28 days after injury (Fig. 2d). Of note, the media area and circumference of the external elastic lamina did not differ between WT and Srsf1$^{-/-}$ mice (Fig. 2d). PCNA immunohistochemical assays showed that SRSF1 deficiency markedly suppressed proliferation as manifested by a profound reduction in the number of PCNA-positive cells (Fig. 2e,f). Taken together, these results indicate that SRSF1 deficiency impairs cell proliferation, thereby suppressing neointima formation.

To test whether SRSF1 is involved in postinjury endothelial recovery, we measured re-endothelialization by en face Evans blue staining of the denuded area at 3 and 7 days after injury (Fig. 2g,h). WT endothelial cells were severely damaged immediately after injury and recovered by 30% on day 3 and 67% on day 7. Endothelial recovery in Srsf1$^{-/-}$ mice was 16% on day 3 and 34% on day 7 postinjury (Fig. 2g,h), suggesting that SRSF1 deletion impairs re-endothelialization.

To validate the vascular function of SRSF1, we further used another inducible SMC-specific *Srsf1*-knockout mouse strain, which was generated by crossing floxed *Srsf1* mice with *SMA-Cre^{ERT2}* transgenic mice (*Srsf1^{flox/flox}* × *SMA-Cre^{ERT2}*), in which the smooth muscle actin promoter drives the expression of the tamoxifen-inducible Cre^{ERT2} recombinase (Fig. 3a). The SRSF1 expression was specifically downregulated in the aorta of *Srsf1^{−/−}*/Cre^{ERT2} mice (*Srsf1^{flox/flox}*; *SMA-Cre^{ERT2+}*) compared with WT littermate controls (*Srsf1^{flox/flox}*; *SMA-Cre^{ERT2−}*)

(Fig. 3b). Under baseline conditions at 10–12 weeks of age, there was no difference in blood pressure or heart rate between *Srsf1^{−/−}*/Cre^{ERT2} mice and their WT littermates (Supplementary Fig. 3d,e). Tamoxifen-induced deletion of SRSF1 overtly attenuated the wire injury-induced neointima formation (Fig. 3c,d). On day 14 postinjury, the average intima/media ratio in *Srsf1^{−/−}*/Cre^{ERT2} mice was very low and on day 28 decreased to 21.6% of the value in the WT group (Fig. 3d). Meanwhile, the media size did not differ between the two groups (Fig. 3d).

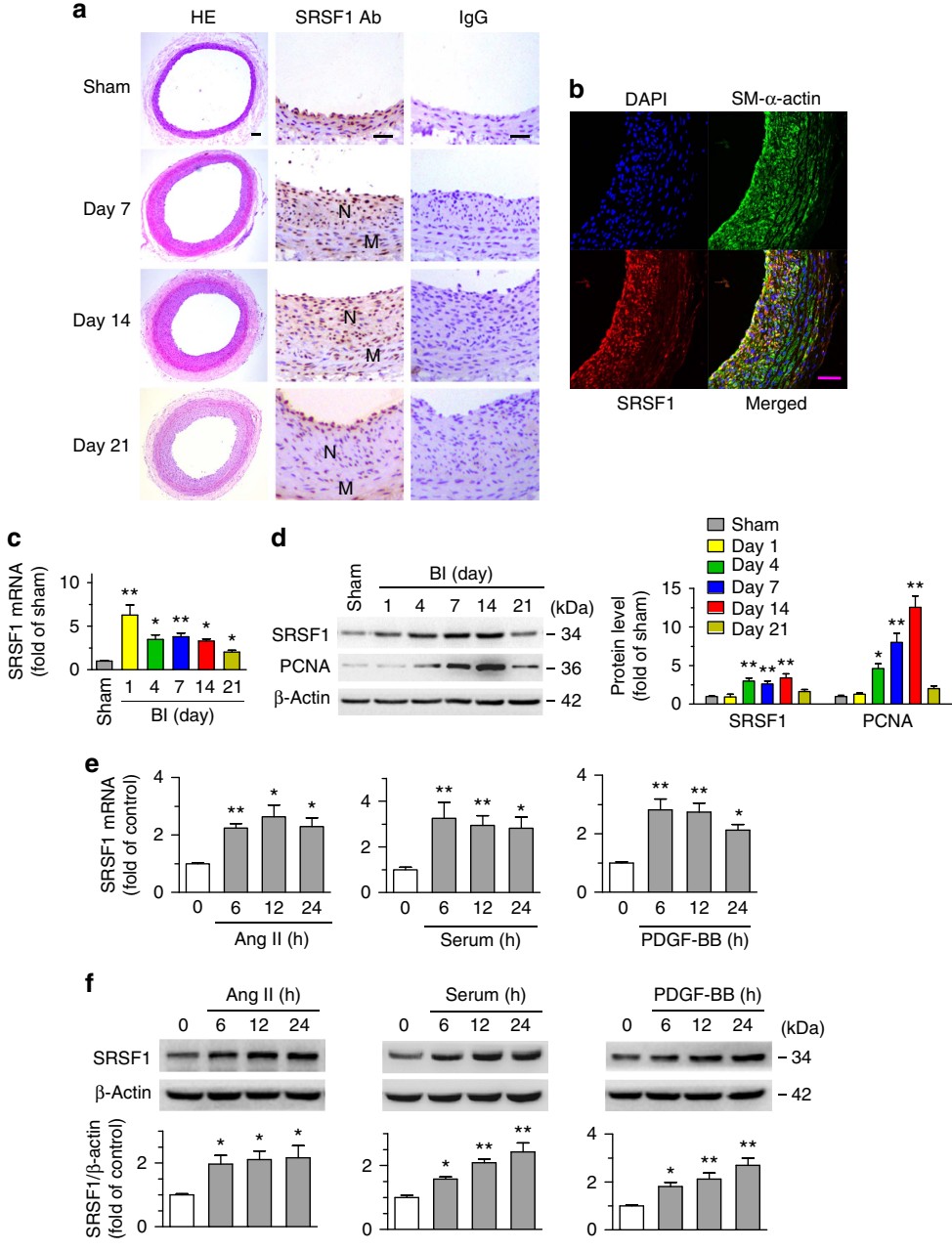

**Figure 1 | Increased SRSF1 expression in proliferating VSMCs *in vivo* and *in vitro*.** (**a**) Photomicrographs of haematoxylin/eosin-stained carotid arteries from sham-operated and balloon-injured rats. Immunohistochemical staining of vessels with specific anti-SRSF1 antibody revealed SRSF1 mainly in the neointima (N, neointima; M, media). Normal rabbit IgG served as a negative control. Scale bars, 50 μm. (**b**) Immunofluorescence double staining of injured carotid arteries with specific antibodies against SM-α-actin (green) or SRSF1 (red). Nuclei were stained with DAPI (blue), and yellow indicates their co-localization in the merged images. Scale bars, 50 μm. (**c**) Real-time PCR showing the mRNA levels of SRSF1 in carotid arteries at 1, 4, 7, 14 and 21 days after balloon injury; $n = 8$ per group. (**d**) Representative western blots and averaged data showing SRSF1 and PCNA levels in rat carotid arteries at 1, 4, 7, 14 and 21 days after balloon injury; $n = 8$ per group. (**e**) Real-time PCR data showing the mRNA levels of SRSF1 in HASMCs treated with Ang II (200 nM), serum (10% FBS) or PDGF-BB (10 μg l$^{-1}$) at 6, 12 and 24 h; $n = 8$ per group. (**f**) Representative western blots and averaged data showing SRSF1 levels in HASMCs treated as in **e**; $n = 7$ per group. *$P < 0.05$, **$P < 0.01$, one-way ANOVA (**c–f**). Data are mean ± s.e.m. of five independent experiments (**c–f**).

PCNA-positive cells were fewer in $Srsf1^{-/-}$/$Cre^{ERT2}$ mice at 14 and 28 days after injury (Fig. 3e,f). Similar to $Srsf1^{-/-}$ mice, re-endothelialization was suppressed in $Srsf1^{-/-}$/$Cre^{ERT2}$ mice:

17% on day 3, and 33% on day 7 postinjury (Fig. 3g,h). These results further support the idea that SRSF1 deficiency suppresses neointima formation.

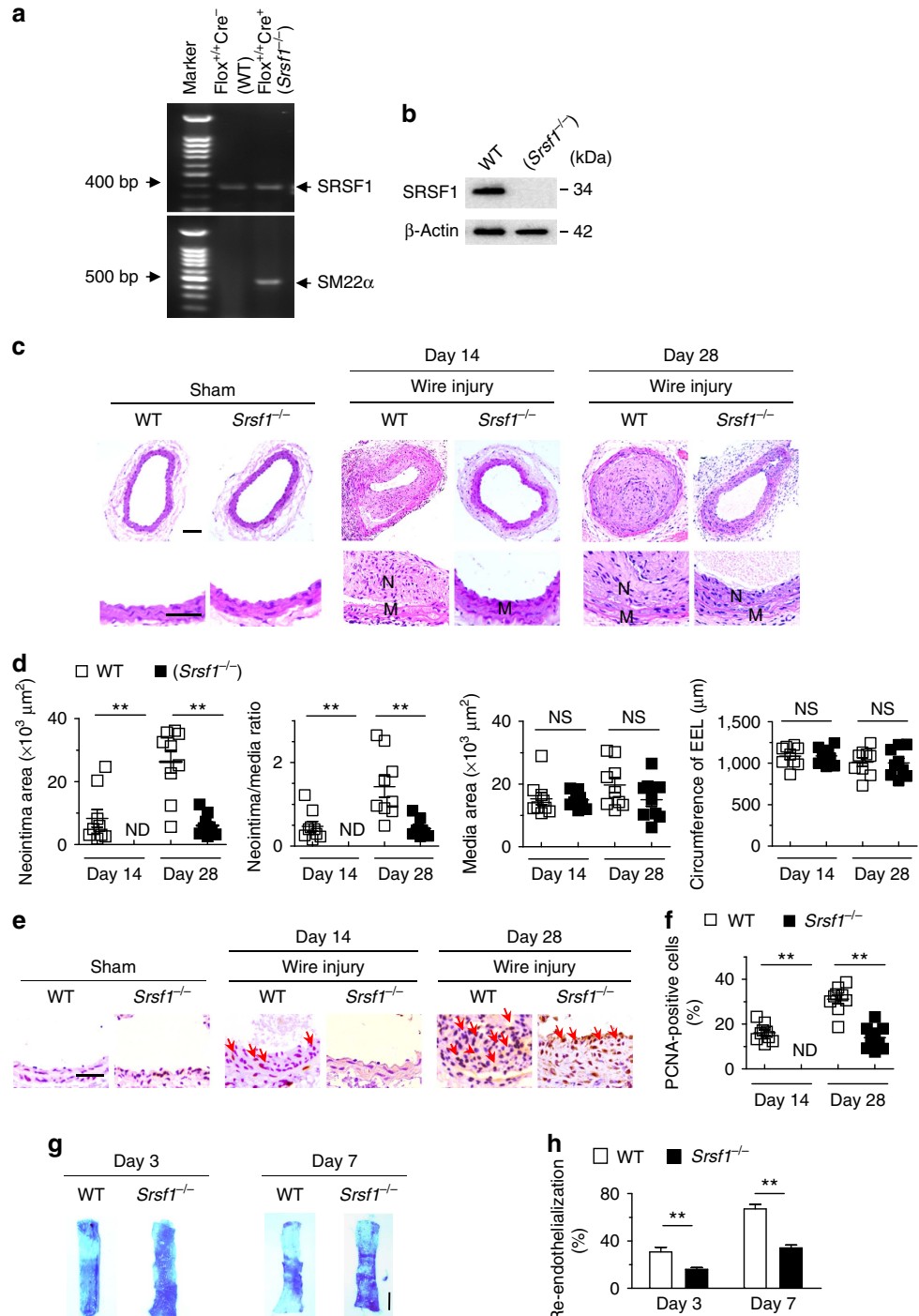

**Figure 2 | SRSF1 deficiency in VSMCs inhibits neointima formation.** (**a**) PCR showing genotyping of WT (Flox$^{+/+}$Cre$^-$) and $Srsf1^{-/-}$ (Flox$^{+/+}$Cre$^+$) mice. Band in the upper image indicates the product from SRSF1$^{flox/flox}$, and lower image indicates product from SM22α-Cre. (**b**) Representative western blots showing SRSF1 levels in vessel from smooth muscle cell-specific $Srsf1$ knockout ($Srsf1^{-/-}$) mice. (**c,d**) Representative photomicrographs of haematoxylin and eosin staining (scale bars, 50 μm) (**c**) and averaged data (**d**) of the neointimal area, neointima/media ratio, media area and circumference of external elastic lamina (EEL) of carotid arteries from $Srsf1^{-/-}$ and WT control mice 14 and 28 days after wire injury (N, neointima; M, media); $n = 9$ per group. (**e,f**) Representative photomicrographs of immunohistochemical staining (scale bar, 25 μm) (**e**) and averaged data (**f**) showing the percentages of PCNA-positive cells in carotid arteries from $Srsf1^{-/-}$ and WT control mice 14 and 28 days after wire injury. Arrows indicate PCNA-positive cells (dark brown); $n = 9$ per group. (**g,h**) Representative pictures (**g**) and averaged data (**h**) of re-endothelialization. Re-endothelialization was quantified in Evans blue-stained carotid arteries at 3 and 7 days after vascular injury. Blue staining indicates endothelial denudation. Scale bar, 1 mm; $n = 9$ per group. **$P < 0.01$, NS, not significant; Student's $t$-test (**d,f,h**). Data are mean ± s.e.m. of five independent experiments (**d,f,h**).

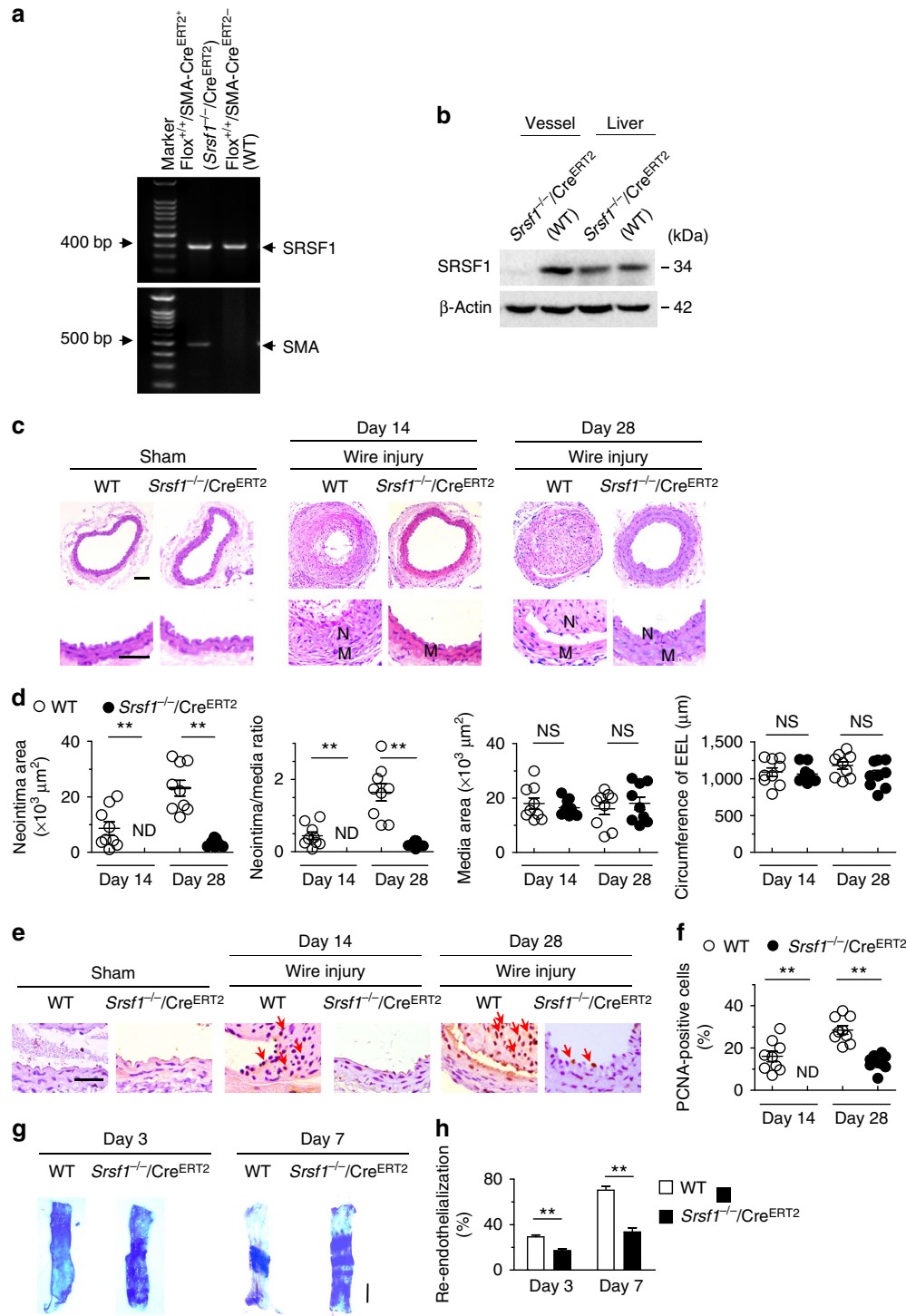

**Figure 3 | Inducible SMC-specific SRSF1 deficiency inhibits neointima formation.** (**a**) PCR showing genotyping of WT (Flox$^{+/+}$Cre$^{ERT2-}$) and
$Srsf1^{-/-}$/Cre$^{ERT2}$ (Flox$^{+/+}$Cre$^{ERT2+}$) mice. Band in the upper image indicates the product from SRSF1$^{flox/flox}$, and lower image indicates product from
SMA-Cre$^{ERT2}$. (**b**) Representative western blots showing SRSF1 levels in the vessel and liver from inducible smooth muscle cell-specific $Srsf1$ knockout
($Srsf1^{-/-}$/Cre$^{ERT2}$) mice. (**c,d**) Representative photomicrographs of haematoxylin and eosin staining (scale bars, 50 μm) (**c**) and averaged data (**d**) of the
neointimal area, neointima/media ratio, media area and circumference of external elastic lamina (EEL) of carotid arteries from $Srsf1^{-/-}$/Cre$^{ERT2}$ and WT
control mice 14 and 28 days after wire injury (N, neointima; M, media); $n = 9$ per group. (**e,f**) Representative photomicrographs of immunohistochemical
staining (scale bar, 25 μm) (**e**) and averaged data (**f**) showing the percentages of PCNA-positive cells in carotid arteries from $Srsf1^{-/-}$/Cre$^{ERT2}$ and WT
control mice 14 and 28 days after wire injury. Arrows indicate PCNA-positive cells (dark brown); $n = 9$ per group. (**g,h**) Representative pictures (**g**) and
averaged data (**h**) of re-endothelialization. Re-endothelialization was quantified in Evans blue-stained carotid arteries at 3 and 7 days after vascular injury.
Blue staining indicates endothelial denudation. Scale bar, 1 mm; $n = 9$ per group. **$P < 0.01$, NS, not significant; Student's $t$-test (**d,f,h**). Data are
mean ± s.e.m. of five independent experiments (**d,f,h**).

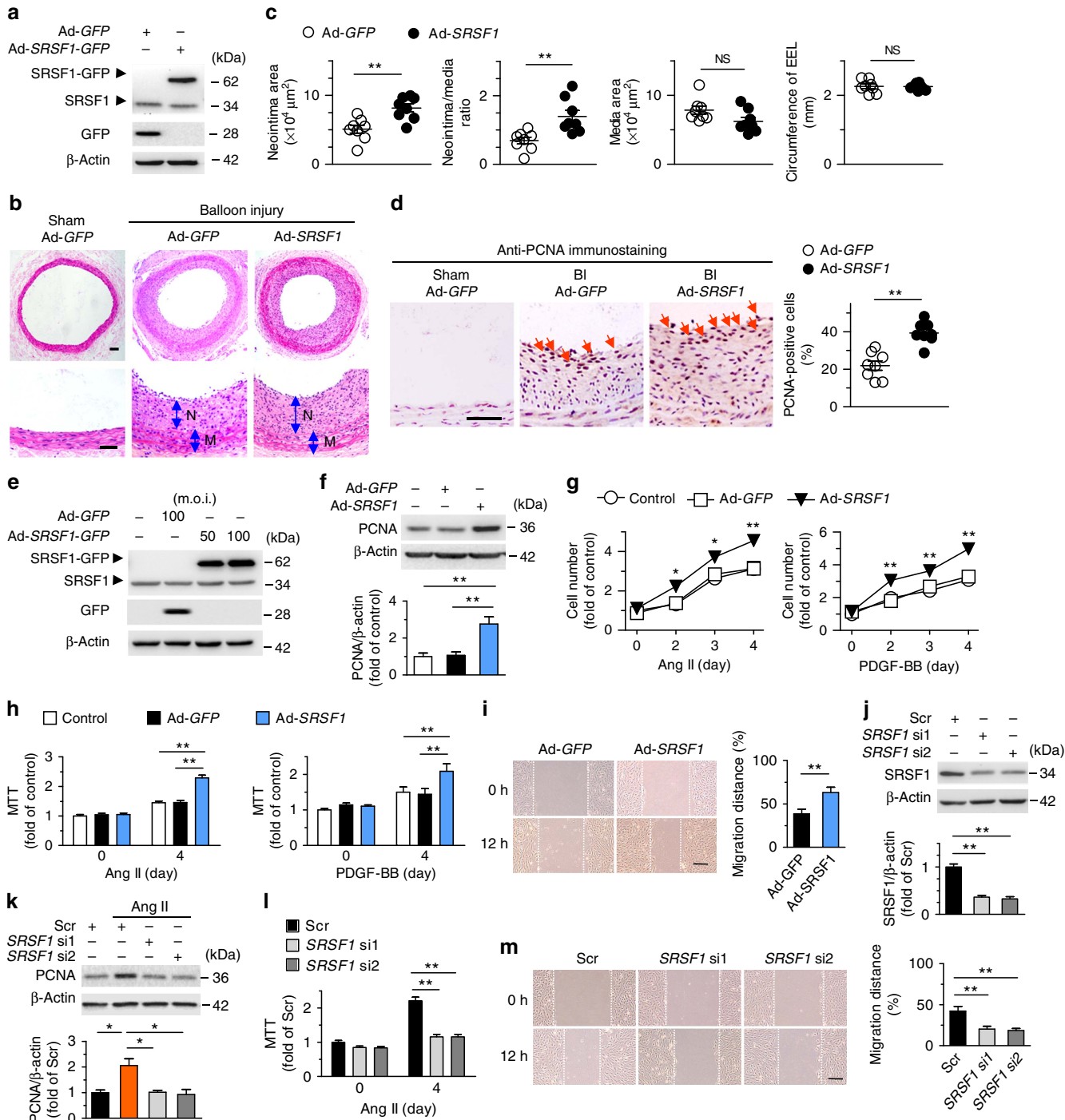

**Figure 4 | SRSF1 enhances HASMC proliferation and neointima formation. (a)** Western blots showing SRSF1 protein in rat carotid arteries 4 days after Ad-*SRSF1-GFP* delivery; $n = 5$. **(b,c)** Haematoxylin/eosin staining (scale bars, 50 μm) **(b)** and averaged data **(c)** of the neointimal area, neointima/media ratio, media area and circumference of external elastic lamina (EEL) of rat carotid arteries transfected with Ad-*GFP* or Ad-*SRSF1* 14 days postinjury (N, neointima; M, media); $n = 8$ each. **(d)** PCNA staining (scale bar, 25 μm) (left) and the percentage of PCNA-positive cells (right) in rat carotid arteries infected with Ad-*SRSF1* 14 days postinjury. Arrows indicate PCNA-positive cells (dark brown); $n = 8$ each. **(e)** SRSF1 expression in cultured HASMCs infected with Ad-*GFP* or Ad-*SRSF1* (m.o.i. 50 and 100); $n = 5$. **(f)** PCNA protein levels in HASMCs infected with Ad-*GFP* or Ad-*SRSF1*; $n = 5$ each. **(g,h)** Cell counts **(g)** and MTT assays **(h)** of HASMCs infected with Ad-*GFP* or Ad-*SRSF1* after Ang II or PDGF-BB stimulation for 4 days; $n = 12$ each. **(i)** Representative images and migration distance from wound-healing assays in HASMCs infected with Ad-*GFP* or Ad-*SRSF1*; $n = 9$ each. Scale bar, 200 μm. **(j)** SRSF1 expression in HASMCs infected with scrambled or *SRSF1* siRNAs (si1 and si2); $n = 6$ each. **(k)** PCNA levels in cultured HASMCs infected with scrambled or *SRSF1* siRNAs after Ang II stimulation (24 h); $n = 5$ each. **(l)** MTT assays of HASMCs infected with *SRSF1* siRNAs after Ang II stimulation for 4 days; $n = 9$ each. **(m)** Representative images and averaged data from wound-healing assays in HASMCs infected with *SRSF1* siRNAs and treated with Ang II; $n = 12$ each; scale bar, 200 μm. All adenoviral infection above is 100 m.o.i. for 48 h unless specified. Ang II is 200 nM and PDGF-BB is 10 μg l$^{-1}$. Scr indicates scrambled siRNA control. *$P < 0.05$, **$P < 0.01$, NS, not significant; Student's *t*-test **(c,d,i)** or one-way ANOVA **(f,h,j–m)** or two-way ANOVA **(g)**. Data are mean ± s.e.m. of five **(c,d,f,j,k)** or four **(g–i,l,m)** independent experiments.

**SRSF1 promotes VSMC proliferation and neointimal thickening.** We next determined whether SRSF1 promotes neointima formation. Rat carotid arteries were subjected to balloon injury and simultaneously infected with either Ad-*GFP* or Ad-*SRSF1*-GFP, as described previously[19,20]. The efficiency of *in vivo* adenoviral gene transfer of SRSF1 was confirmed by western blot 4 days after infection (Fig. 4a). Forced expression of SRSF1 enhanced the neointima formation induced by balloon injury (Fig. 4b). The arterial neointimal area (Fig. 4c) and neointima/media ratio (Fig. 4c) were higher in the Ad-*SRSF1* group than in the Ad-*GFP* group, while no difference was noted in the media area and circumference of the external elastic lamina (Fig. 4c). Overexpression of SRSF1 increased the number of proliferating PCNA-positive cells (Fig. 4d).

Proliferation and migration of VSMCs are key processes in neointima formation in response to arterial injury or mitogenic factors, so we investigated the potential effect of SRSF1 on both processes in the presence of Ang II or PDGF-BB. Here we enhanced or suppressed the SRSF1 expression in cultured HASMCs. PCNA protein was increased by Ad-*SRSF1* (Fig. 4e,f), which also enhanced the Ang II-induced increase of HASMC number in a time-dependent manner compared with that of untreated controls or Ad-*GFP* (Fig. 4g); this enhanced proliferation was confirmed by 3-[4,5-dimethylthiazol-2-yl]-2,5 diphenyl tetrazolium bromide (MTT) assays (Fig. 4h). Likewise, SRSF1 overexpression exaggerated the PDGF-BB-evoked HASMC proliferation as assayed by cell counts and MTT assays (Fig. 4g,h). The wound-healing assay revealed that Ad-*SRSF1* enhanced the Ang II-stimulated VSMC migration (Fig. 4i). These results suggest that upregulation of SRSF1 is sufficient to stimulate HASMC proliferation and migration.

The endogenous expression of SRSF1 in HASMCs was efficiently downregulated by two sets of short interfering RNAs (siRNAs; Fig. 4j, Supplementary Fig. 18). PCNA protein, which was upregulated by Ang II, was attenuated by *SRSF1* siRNAs (Fig. 4k). Knockdown of SRSF1 markedly abrogated Ang II-triggered HASMC proliferation as measured in MTT assays (Fig. 4l) and reduced Ang II-induced migration as measured by the wound-healing assay (Fig. 4m), substantiating the proliferation- and migration-enhancing effects of SRSF1 on HASMCs.

**SRSF1 modulates Δ133p53 in regulating VSMC proliferation.** Since p53 is an important regulator of VSMC growth[20] and p53 signals are a primary response to SRSF1 in human and mouse fibroblasts[3,21], we assessed the p53 expression in SRSF1-overexpressing HASMCs. Surprisingly, we found that, in HASMCs, Ad-*SRSF1* upregulated Δ133p53, a short isoform of p53 (Fig. 5a, Supplementary Fig. 19), but did not alter full-length p53 and its other isoforms (Supplementary Fig. 4), as analyzed by p53 isoform-specific reverse transcriptase–polymerase chain reaction (PCR)[22]. Given there are three variants (α, β and γ) of Δ133p53, we verified that it was the α variant (Δ133p53α) being detected, called Δ133p53 in this study (Supplementary Fig. 5). Western blots revealed an increase of Δ133p53 protein in HASMCs overexpressing SRSF1 (Fig. 5b), but a decrease in those depleted of SRSF1 (Fig. 5c,d). One mouse isoform is highly homologous to human Δ133p53 (Supplementary Fig. 6). This isoform has been reported as mouse Δ157p53[23] and we have named it MΔ133p53 here. We found that MΔ133p53 was lower in the aortas from SRSF1-deficient mice than in WT controls (Fig. 5e). Moreover, Δ133p53 responded to proliferative stimuli in a way similar to SRSF1, in that treating HASMCs with Ang II (200 nM), serum (10% FBS) or PDGF-BB (10 µg l$^{-1}$) enhanced Δ133p53 expression at the mRNA and protein levels in a time-dependent manner (Supplementary Fig. 7). These data suggest that SRSF1 induces Δ133p53 expression, which may be a generalized VSMC response to proliferative stimuli.

To explore the role of Δ133p53 in vascular status, we first overexpressed it using adenovirus-mediated gene transfer. Administration of Ad-*Δ133p53* to HASMCs effectively increased the expression of Δ133p53 (Fig. 5f) and the protein level of PCNA (Fig. 5g). Ad-*Δ133p53* enhanced the Ang II-induced HASMC proliferation and PDGF-BB-evoked cell proliferation according to cell counts and MTT assays (Fig. 5h,i). Ang II-induced migration was also facilitated in Δ133p53-overexpressing HASMCs (Fig. 5j). To investigate the role of Δ133p53 in vascular hyperplasia *in vivo*, we assessed the neointima formation in rat carotid arteries with balloon injury and infected with either Ad-*GFP* or Ad-*Δ133p53*. Western blots confirmed the efficiency of Ad-*Δ133p53* infection (Supplementary Fig. 8a). Ad-*Δ133p53* infection resulted in increased injury-induced neointima formation, with larger neointimal area and higher neointima-to-media ratio but unaltered media area compared with the Ad-*GFP* group (Fig. 6a,b). Ad-*Δ133p53* also increased the number of neointimal PCNA-positive cells (Fig. 6c), supporting an activating role of Δ133p53 in VSMC proliferation. On the contrary, p53 had an inhibitory effect on VSMC proliferation, migration and neointima formation (Fig. 5h–j, Supplementary Fig. 8b,c), which has been well studied[20,24,25] and served as a control for Δ133p53 here. Moreover, Δ40p53 recapitulated the phenotype of p53 (Fig. 5h–j, Supplementary Fig. 8b,d). On the other hand, we carried out siRNA silencing to deplete endogenous Δ133p53 (Fig. 6d). *Δ133p53* siRNAs attenuated Ang II-induced PCNA increase (Fig. 6e) and markedly diminished the Ang II-triggered HASMC proliferation (Fig. 6f) and migration (Fig. 6g), substantiating the growth-activating and pro-migratory effects of Δ133p53 on VSMCs.

The upregulation of Δ133p53 by SRSF1 and their equivalent pro-proliferative effects noted above raised the possibility that Δ133p53 functions as a mediator of SRSF1-stimulated cell proliferation. Indeed, Δ133p53 is essentially involved, since the proliferation-enhancing effect of SRSF1 was attenuated by *Δ133p53* silencing in HASMCs, as indicated by decreased PCNA abundance and reduced values in MTT assays (Fig. 6h,i). Δ133p53 also participated in migration, as *Δ133p53* silencing retarded SRSF1-induced HASMC migration (Fig. 6j).

**SRSF1 enhances VSMC proliferation via KLF5 signalling.** To delineate the mechanism underlying SRSF1/Δ133p53-mediated changes in VSMC proliferation, we analyzed the genome-wide expression profiles of HASMCs with Δ133p53 overexpression and Ang II treatment versus green fluorescent protein (GFP) controls using an RNA deep-sequencing (RNA-Seq) approach. We found 247 genes that fit the criteria of $> 0.5$ FPKM (fragments per kilobase of exon per million fragments mapped) and an adjusted *P*-value of $< 0.05$ (Wald tests implemented in the DESeq[26] R package) that were commonly upregulated by the overexpression of Δ133p53 and Ang II treatment (Fig. 7a). Importantly, among the 39 upregulated genes associated with developmental biology and signal transduction annotated in the Reactome knowledgebase[27] (Supplementary Table 1), we identified an important transcriptional factor, KLF5 (also known as BETB2).

We first confirmed the regulatory relationship between KLF5 and Δ133p53. The mRNA level and protein abundance of KLF5 were upregulated by Δ133p53 overexpression and downregulated by Δ133p53 knockdown (Fig. 7b,c). Then we hypothesized that KLF5 might be a downstream effector of SRSF1. Remarkably, Ad-*SRSF1* increased KLF5, while *SRSF1* siRNA decreased it at both the mRNA and protein levels (Fig. 7d,e). It is

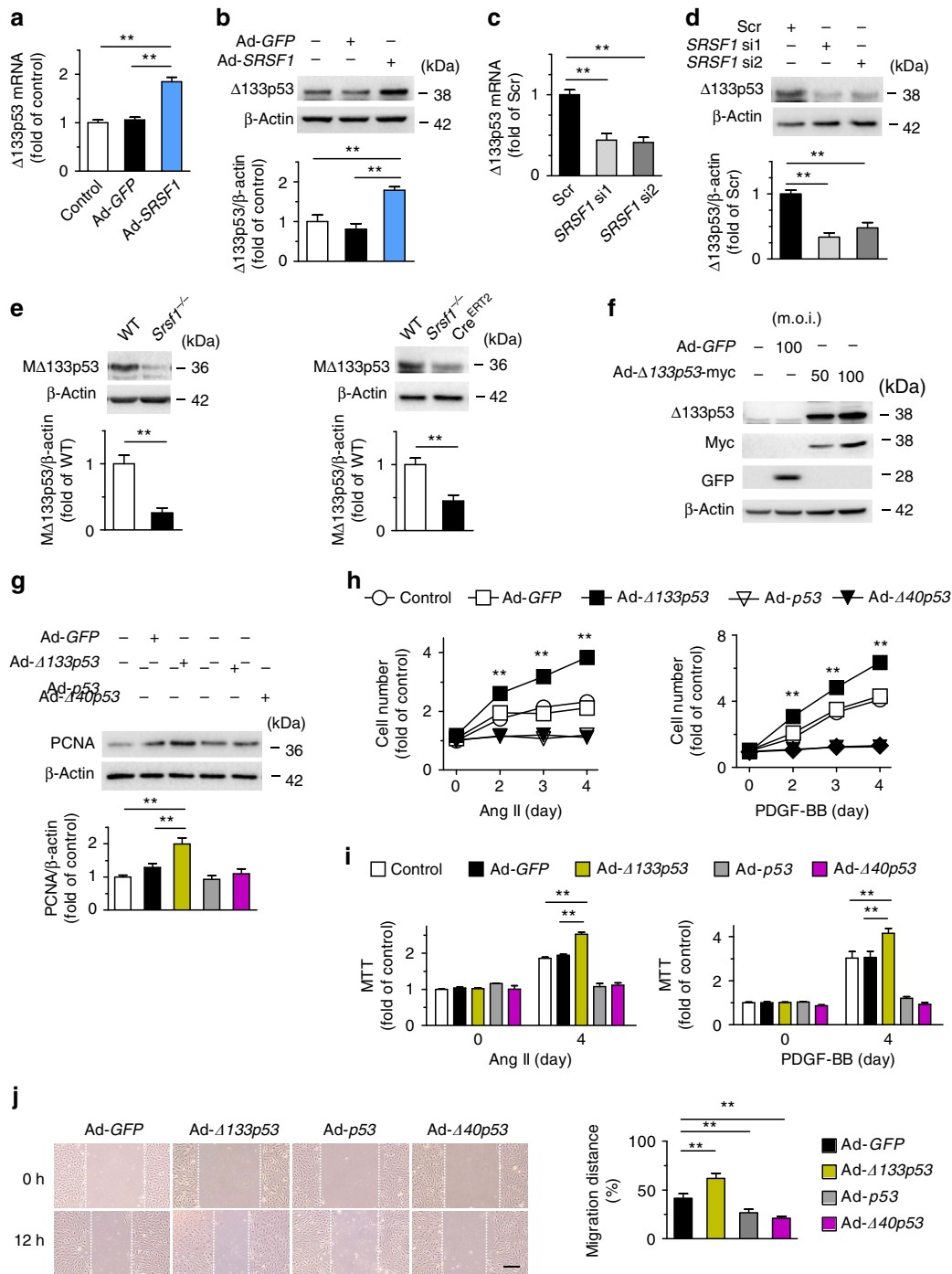

**Figure 5 | SRSF1 modulates Δ133p53 in regulating VSMC proliferation.** (**a,b**) Δ133p53 expression increases by Ad-*SRSF1* infection in HASMCs at mRNA (**a**) and protein (**b**) level; $n = 7$ each. (**c,d**) SiRNA-deletion of SRSF1 reduces mRNA (**c**) and protein (**d**) abundance of Δ133p53; $n = 5$–6 each. (**e**) MΔ133p53 levels in vessels from *Srsf1*$^{-/-}$ mice at 56 days after birth; $n = 8$ each. (**f**) Δ133p53 expression in cultured HASMCs by Ad-*Δ133p53*; $n = 5$ each. (**g–i**) In HASMCs infected with Ad-*GFP*, Ad-*Δ133p53*, Ad-*p53* or Ad-*Δ40p53*, PCNA expression (**g**), cell counts (**h**) and MTT assays (**i**) after Ang II or PDGF-BB stimulation for 4 days are examined; (**g**), $n = 5$; (**h,i**), $n = 12$. (**j**) Wound-healing assays in HASMCs infected with indicated adenovirus and treated Ang II; $n = 9$ each; scale bar, 200 μm. All adenoviral infection above is 100 m.o.i. for 48 h unless specified. Ang II is 200 nM and PDGF-BB is 10 μg l$^{-1}$. **\*\*$P < 0.01$; one-way ANOVA (**a–d,g,i,j**), Student's *t*-test (**e**), or two-way ANOVA (**h**). Data are mean ± s.e.m. of five (**a–d,g**) or four (**e,h–j**) independent experiments.

noteworthy that elevation of SRSF1 and Δ133p53 was universally accompanied by an increase in KLF5 expression in HASMCs exposed to Ang II, serum or PDGF-BB (Fig. 1e,f; Supplementary Figs 7 and 9).

Since enhanced cell-cycle progression contributes directly to proliferation, we explored the mechanism underlying SRSF1-induced cell growth by examining the cell-cycle distribution of HASMCs. Overexpression of SRSF1 increased the percentage of proliferating cells in S and G2/M, with a concomitant decline in the percentage of cells in G0/G1 (Fig. 7f). Moreover, *KLF5*-silencing abolished the SRSF1-induced accumulation of cells in S and G2/M (Fig. 7f,g). These findings suggest that SRSF1 enhances HASMC proliferation by driving cell-cycle progression via a KLF5-dependent pathway.

Thus we further investigated the downstream pathway of KLF5 in terms of cell-cycle regulation. Among the primary target genes of KLF5, p21 is a major cyclin kinase-dependent regulator capable of arresting cell-cycle progression. Consistent with previous studies[28,29], KLF5 decreased the expression of p21 in HASMCs (Supplementary Fig. 10). Overexpression of SRSF1 or Δ133p53 decreased p21 expression (Fig. 7h; Supplementary Fig. 11), while p21 was reciprocally upregulated by siRNA depletion of SRSF1 or Δ133p53 (Fig. 7i; Supplementary Fig. 12), indicating that p21 is a downstream effector of SRSF1 and Δ133p53. KLF5 siRNA depletion attenuated the decrease in p21 and the increase in PCNA caused by SRSF1 overexpression (Fig. 7j), which indicated that KLF5 mediates SRSF1-p21 regulation. Moreover, we found that the SRSF1-induced upregulation of KLF5, as well as the suppression of p21, were attenuated by siRNA against Δ133p53 (Fig. 7k), substantiating the idea that activation of KLF5 signalling by SRSF1 is mediated by Δ133p53 in HASMCs. We further investigated this KLF5–p21 signal in proliferating HASMCs treated with Ang II and found that siRNA depletion of either SRSF1 or Δ133p53 circumvented the Ang II-induced upregulation of KLF5 and abolished the Ang II-induced suppression of p21 (Fig. 7l,m). These results indicate that an SRSF1–Δ133p53–KLF5–p21 signalling pathway is activated in SRSF1-induced cell proliferation.

We next investigated the KLF5 and p21 signals in vivo in Srsf1$^{-/-}$ mice and balloon-injured rat arteries. The expression of KLF5 was lower, while that of p21 was higher in the aortas of Srsf1$^{-/-}$ mice than in controls (Fig. 7n; Supplementary Fig. 13). In rat carotid arteries overexpressing SRSF1, 1 week after injury the KLF5 expression increased while that of p21 decreased (Fig. 7o; Supplementary Fig. 14). These results indicate that KLF5–p21 signalling is also essentially involved in SRSF1-mediated neointima formation.

**KLF5 activation by SRSF1 requires EGR1-Δ133p53 binding**. Previous studies have shown that EGR1 is a critical transcription factor of KLF5 (ref. 30). A physical interaction between Δ133p53 and EGR1 was revealed by co-immunoprecipitation assays (Fig. 8a), while SRSF1 did not interact with EGR1 (Supplementary Fig. 15). Bioinformatics analysis predicted four EGR1-binding domains in the KLF5 promoter, which might be EGR1-responsive sites underlying the transcriptional regulation of KLF5 (Supplementary Fig. 16). Chromatin immunoprecipitation (ChIP)-PCR assays showed that Δ133p53 enhanced EGR1 enrichment to the KLF5 promoter at the p1 and p2 sites (Fig. 8b), and gene silencing of EGR1 attenuated this enrichment (Fig. 8c,d), indicating that Δ133p53 transcriptionally regulates KLF5 at the p1 and p2 sites via activating the transcription factor EGR1. Furthermore, knockdown of EGR1 fully eliminated the upregulation of KLF5 and PCNA, as well as the downregulation of p21, induced by either SRSF1 or Δ133p53 overexpression (Fig. 8e,f). The enhancement of proliferation induced by SRSF1 or Δ133p53 in Ang II-treated HASMCs was also abolished by gene silencing of EGR1 as assessed by MTT assays (Fig. 8g,h). These data suggest that EGR1, through forming the functional complex Δ133p53-EGR1, is obligatory for the SRSF1-induced activation of KLF5 signalling and thereby essential for SRSF1-mediated proliferation.

**SRSF1 blocks apoptosis via activating Bcl-xL**. Given that VSMC apoptosis is closely associated with vascular remodelling[31], we next determined whether apoptosis is involved in the SRSF1-mediated VSMC proliferation. The balloon injury-induced apoptosis was attenuated by Ad-SRSF1 overexpression in rat carotid arteries as assayed by terminal deoxinucleotidyl transferase-mediated dUTP-

fluorescein nick end labelling (TUNEL) staining (Supplementary Fig. 17a). Cell viability assays showed that SRSF1 overexpression rescued H$_2$O$_2$-triggered HASMC death (Supplementary Fig. 17b), and this improved cell survival was attributed to the blockade of H$_2$O$_2$-induced apoptosis by SRSF1, as indicated by decreased DNA laddering and reduced caspase 3 activity (Supplementary Fig. 17c,d). In sharp contrast, knockdown of SRSF1 resulted in compromised cell survival (Supplementary Fig. 17e) due to HASMC apoptosis, as indicated by increased DNA laddering and elevated caspase 3 activity (Supplementary Fig. 17f). Moreover, TUNEL staining showed that the VSMC apoptosis induced by wire injury was markedly higher in Srsf1$^{-/-}$ mice than in WT controls (Supplementary Fig. 17g).

To investigate the candidate mechanism underlying the antiapoptotic effect of SRSF1, we tested the expression of B-cell lymphoma-extra large (Bcl-xL), a powerful antiapoptotic protein. Surprisingly, SRSF1 elevated the Bcl-xL expression in vivo and in vitro (Supplementary Fig. 17h–k), while its expression was lower in the aortas of Srsf1$^{-/-}$ mice (Supplementary Fig. 17l,m) and in SRSF1-silenced HASMCs (Supplementary Fig. 17n,o). In addition, the Bcl-xL induction by SRSF1 was attenuated in the presence of Δ133p53 siRNA (Supplementary Fig. 17p), while Ad-Δ133p53 rescued the inhibition of Bcl-xL by SRSF1 siRNA (Supplementary Fig. 17q), suggesting that Δ133p53 is a mediator of the SRSF1–Bcl-xL signal. Taken together, these data suggest that SRSF1 activates Bcl-xL to block VSMC apoptosis.

**SRSF1 facilitates endothelial cell proliferation**. Healthy vascular endothelial cells secrete antiproliferative and antithrombotic substances to guarantee a regular blood flow. Rapid endothelial recovery, or re-endothelialization, is correlated with diminished neointima formation, while inadequate re-endothelialization may result in thrombosis[32]. Thus we evaluated the effects of SRSF1 and Δ133p53 on re-endothelialization. Endothelial recovery was improved by SRSF1 overexpression on days 3 and 7 after balloon injury (Fig. 9a), which was consistent with the reduced endothelium repair in SRSF1-deficient mice (Figs 2g,h and 3g,h). Δ133p53 also improved re-endothelialization (Fig. 9a).

Because migration and proliferation of endothelial cells are essential processes in re-endothelialization[32], we further explored the effects of SRSF1 on both processes in cultured human coronary artery endothelial cells (HCAECs). SRSF1 expression increased in Ang II-stimulated HCAECs (Fig. 9b,c), and SRSF1 overexpression upregulated Δ133p53 (Fig. 9d,e). An in vitro scratch-wound assay revealed that SRSF1 and Δ133p53 promoted Ang II-triggered HCAEC migration (Fig. 9f,g), and the SRSF1-induced enhancement of migration was attenuated by Δ133p53 knockdown (Fig. 9h,i). Overexpression of SRSF1 and Δ133p53 in HCAECs effectively increased the PCNA abundance (Fig. 9j,k), enhanced the Ang II-induced cell proliferation (Fig. 9l) and the SRSF1-induced HCAEC proliferation was mediated by Δ133p53 (Fig. 9m,n). These data indicate that SRSF1 promotes endothelial cell migration and proliferation via the Δ133p53 signal.

**SRSF1 promotes cell migration and proliferation in HCASMCs**. Small arteries such as coronary and peripheral arteries after angioplasty are prone to atherosclerosis or restenosis due to injury-induced intimal hyperplasia[1]. Thus, to further clarify the relevance of SRSF1 in human physiology and pathology, we examined the role of SRSF1 in human coronary artery SMCs (HCASMCs). SRSF1 increased in response to Ang II stimulation (Fig. 10a,b) and SRSF1 upregulated Δ133p53 in HCASMCs (Fig. 10c,e). Overexpression of SRSF1 and Δ133p53 enhanced the Ang II-induced HCASMC migration, as assayed by wound healing (Fig. 10d,f), and cell proliferation, as measured by PCNA abundance and MTT assay (Fig. 10g–i). SRSF1-enhanced

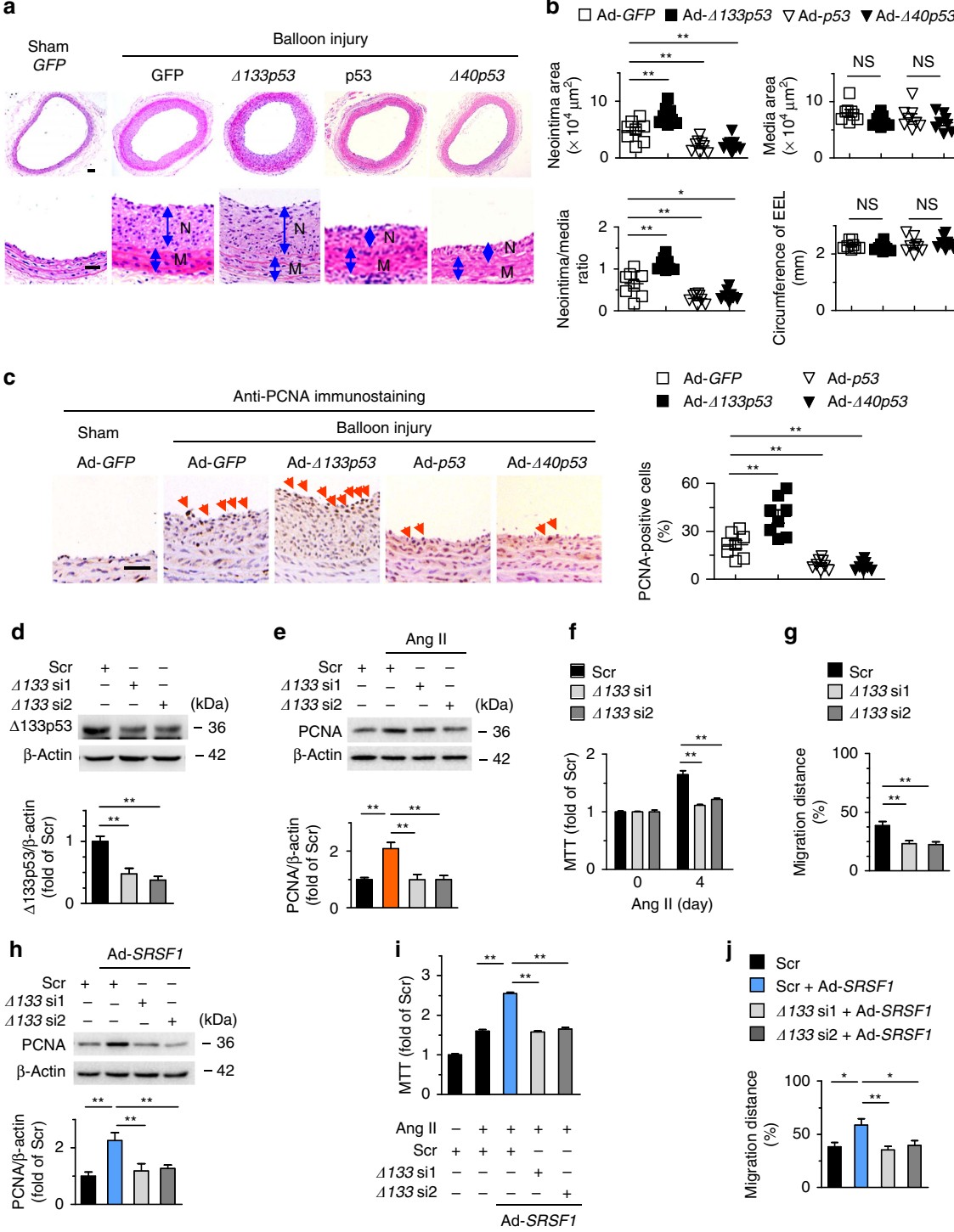

**Figure 6 | Δ133p53 promotes neointima formation after vascular injury.** (**a,b**) Hematoxylin/eosin staining (**a**) and averaged data (**b**) of the neointimal area, neointima/media ratio, media area, and circumference of external elastic lamina (EEL) of rat carotid arteries transfected with indicated adenovirus 2 weeks postinjury (N, neointima; M, media); $n = 8$ each; scale bar, 50 μm. (**c**) PCNA staining and PCNA-positive cells in rat carotid arteries treated as in (**a**) 2 weeks postinjury. Arrows indicate PCNA-positive cells; $n = 8$ each; scale bar, 25 μm. (**d**) Δ133p53 siRNAs (Δ133 si1 and Δ133 si2) knock down Δ133p53 level in HASMCs; $n = 8$. (**e–g**) PCNA level (**e**), MTT assays (**f**) and wound-healing assays (**g**) in HASMCs infected with Δ133p53 siRNAs after Ang II stimulation; (**e**), $n = 8$; (**f**), $n = 11$; (**g**), $n = 9$. (**h**) PCNA levels in HASMCs infected with Δ133p53 siRNAs in the presence or absence of Ad-SRSF1; $n = 6$. (**i**) MTT assays showing SRSF1-mediated enhancement of proliferation was inhibited by Δ133p53 knockdown; $n = 12$. (**j**) Averaged data from wound-healing assays in HASMCs infected with Δ133p53 siRNAs in the presence or absence of Ad-SRSF1 and treated with Ang II; $n = 8$. All adenoviral infection in HASMCs above is 100 m.o.i. for 48 h. Ang II is 200 nM. Scr indicates scrambled siRNA. *$P < 0.05$, **$P < 0.01$, NS, not significant; one-way ANOVA (**b–j**), Data are mean ± s.e.m. of five (**b–e,h**) or four (**f,g,i,j**) independent experiments.

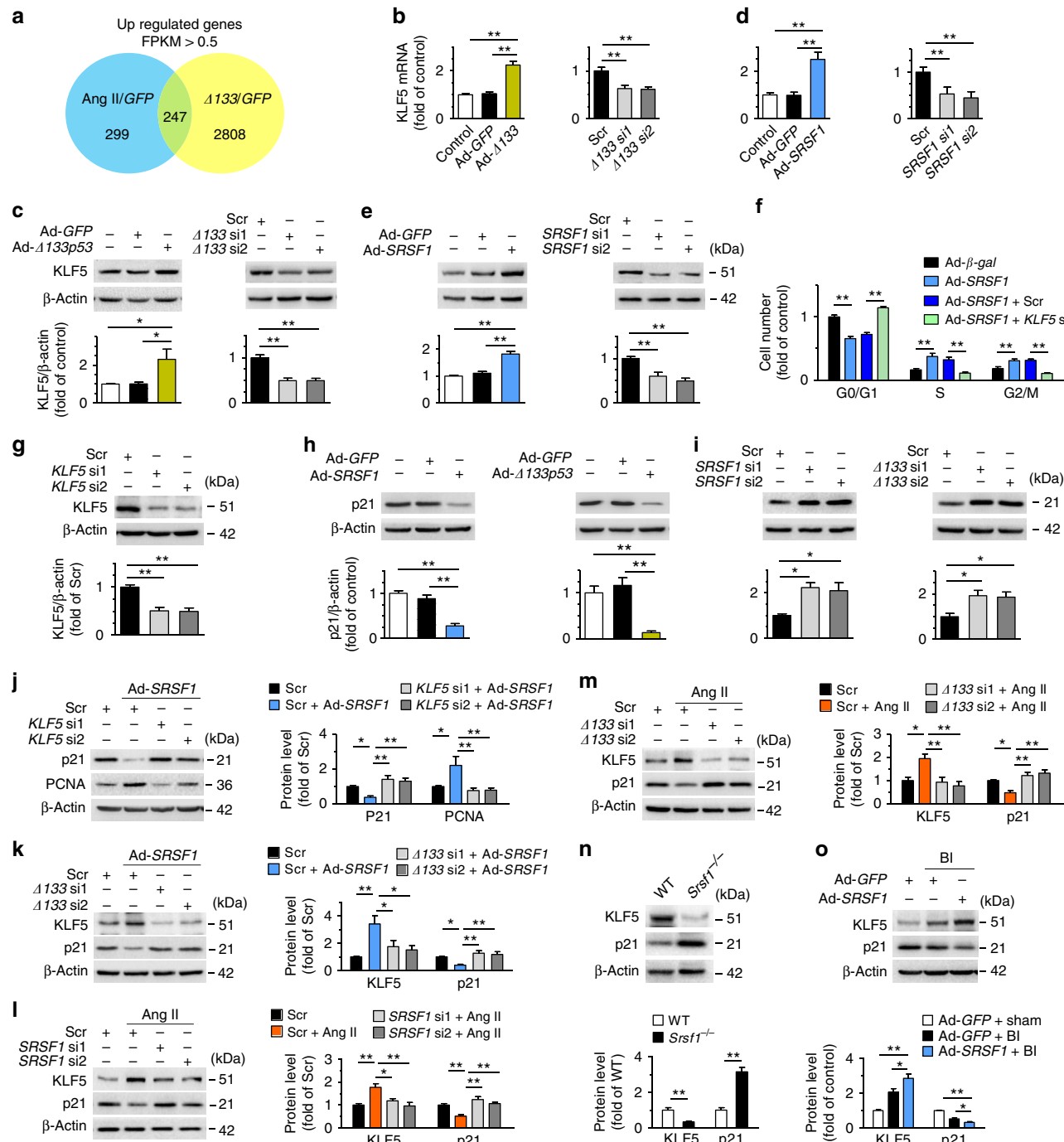

**Figure 7 | SRSF1 promotes VSMC proliferation via KLF5–p21 signal.** (**a**) Venn diagram of modulated genes in HASMCs overexpressing Δ133p53 and treated with Ang II, showing 247 genes that fit the criteria of >0.5 FPKM (fragments per kilobase of exon per million fragments mapped) and an adjusted *P*-value of <0.05 (Wald tests implemented in the DESeq R package), commonly modulated in both conditions. (**b,c**) The mRNA levels (**b**) and protein abundance (**c**) of KLF5 in HASMCs with Δ133p53 overexpression or Δ133p53 knockdown; n = 7–8 per group. (**d,e**) KLF5 expression in HASMCs with SRSF1 overexpression or SRSF1 knockdown at mRNA level (**d**) and protein level (**e**); n = 6 per group. (**f**) Cell-cycle distributions in HASMCs infected with Ad-*SRSF1* in the absence or presence of *KLF5* siRNA stimulated with Ang II (200 nM) for 24 h; n = 9 per group. (**g**) *KLF5* siRNAs (*KLF5* si1 and *KLF5* si2) knock down KLF5 protein in HASMCs; n = 8 per group. (**h**) Expression of p21 in HASMCs infected with Ad-*SRSF1* or Ad-*Δ133p53*; n = 5 per group. (**i**) p21 levels in HASMCs with SRSF1 knockdown or Δ133p53 knockdown; n = 7 per group. (**j**) PCNA and p21 levels in HASMCs infected with *KLF5* siRNAs in the absence or presence of Ad-*SRSF1*; n = 7 per group. (**k**) KLF5 and p21 levels in HASMCs infected with *Δ133p53* siRNAs (*Δ133* si1 and *Δ133* si2) in the presence or absence of Ad-*SRSF1*; n = 7 per group. (**l,m**) KLF5 and p21 levels in cultured HASMCs infected with *SRSF1* siRNAs (**l**) or *Δ133p53* siRNAs (**m**) after Ang II stimulation (200 nM, 24 h); n = 5–7 per group. (**n**) KLF5 and p21 levels in the arteries from *Srsf1*$^{-/-}$ or WT control mice; n = 7 per group. (**o**) KLF5 and p21 levels in rat carotid arteries transfected with Ad-*SRSF1* 1 week after balloon injury; n = 10 per group. Scr indicates scrambled siRNA control; BI, balloon injury. *P < 0.05, **P < 0.01, one-way ANOVA (**b–f,h–m,o**) or Student's *t*-test (**n**). Data are mean ± s.e.m. of four (**b,d**) or five (**c,e–o**) independent experiments.

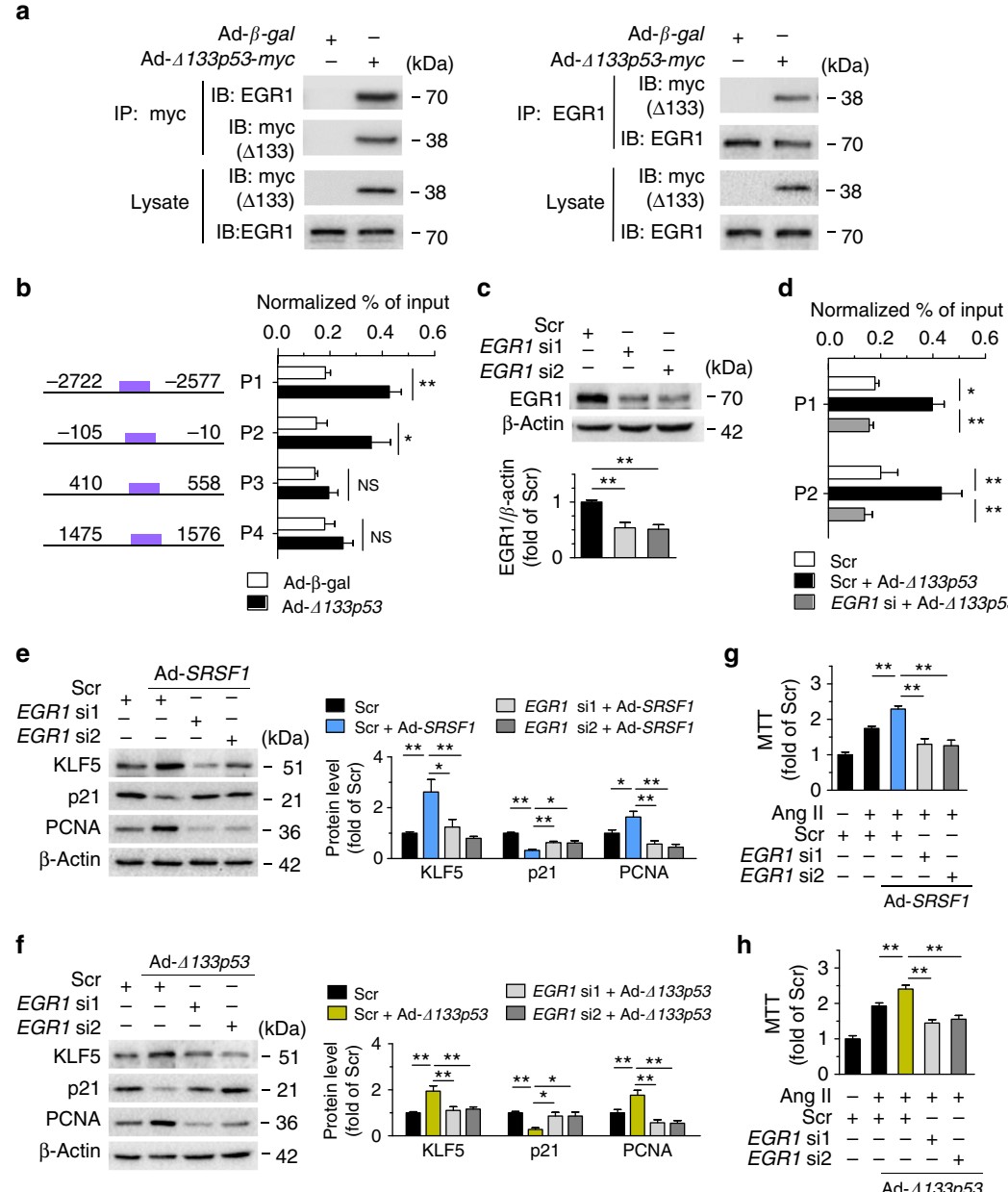

**Figure 8 | SRSF1-triggered activation of KLF5 requires EGR1 binding to Δ133p53. (a)** Co-immunoprecipitation (IP) assay identified interaction between Δ133p53 and EGR1. Cell lysates from HASMCs infected with Ad-β-gal or Ad-Δ133p53-myc were immunoprecipitated with antibody to myc (left) or EGR1 (right) and detected with western blots. Input lysates were loaded as control; $n = 5$ per group. **(b)** Chromatin immunoprecipitation (ChIP) assays were performed to detect the recruitment of EGR1 protein to the promoter of KLF5 after overexpression of Δ133p53. HASMCs were infected with Ad-β-gal or Ad-Δ133p53-myc. Crosslinked DNA–protein lysates are immunoprecipitated with antibody to EGR1. The fragments containing consensus EGR1-binding motifs are PCR amplified from both samples. Four distinct EGR1-binding positions (P1, P2, P3 and P4) in KLF5 are shown. $n = 9$ per group. **(c)** Representative western blots and averaged data showing the EGR1 levels in HASMCs infected with scrambled or two sets of *EGR1* siRNAs (*EGR1* si1 and *EGR1* si2). **(d)** ChIP assays were performed in HASMCs infected with Ad-β-gal or Ad-Δ133p53-myc in the presence or absence of *EGR1* siRNA1 (*EGR1* si); $n = 9$ per group. Crosslinked DNA–protein lysates from infected cells are immunoprecipitated with EGR1 antibody. The P1 and P2 promoter fragments (same as Fig. 7b) containing EGR1-binding motifs are PCR amplified from both samples. **(e)** Representative western blots and averaged data showing KLF5, p21 and PCNA levels in HASMCs transfected with *EGR1* siRNAs with or without Ad-*SRSF1* overexpression; $n = 7$ per group. **(f)** Representative western blots and averaged data showing KLF5, p21 and PCNA levels in HASMCs transfected with *EGR1* siRNAs with or without Ad-Δ133p53 overexpression; $n = 9$ per group. **(g,h)** The proliferation enhancement induced by SRSF1 (**g**) or Δ133p53 (**h**) was attenuated by *EGR1* siRNAs; $n = 12$ per group. Scr indicates scrambled siRNA control. *$P < 0.05$, **$P < 0.01$, NS, not significant; Student's *t*-test (**b**) or one-way ANOVA (**d–h**). Data are mean ± s.e.m. of four (**b,d**) or five (**e–h**) independent experiments.

migration and proliferation was attenuated by Δ133p53 knockdown (Fig. 10j–m). Furthermore, SRSF1 activated KLF5 and suppressed p21 through Δ133p53 in HCASMCs (Fig. 10j).

Collectively, consistent with the findings in HASMCs, SRSF1 promotes the migration and proliferation of SMCs from human coronary artery via activating Δ133p53/KLF5 signals.

## Discussion

Abnormal VSMC proliferation is closely associated with neointima formation during atherosclerosis and postinjury restenosis[1]. In the present study, we uncovered an 'SRSF1–Δ133p53–KLF5' axis in VSMCs that mediates neointima formation after vascular injury. Our results showed that SRSF1 was markedly induced in VSMCs by multiple mitogenic stimuli in vitro and in vivo. SRSF1 deficiency blocked the injury-induced neointima formation. RNA-seq analysis revealed that the KLF5 pathway was activated both in Ang II-treated and Δ133p53-overexpressing HASMCs. We further demonstrated that SRSF1 upregulated KLF5 expression via a Δ133p53–EGR1 complex. By this means, SRSF1 stimulated SMC proliferation and neointima formation (Fig. 10n).

Although originally identified as a splicing factor[5,33], the archetypal SR protein SRSF1 has been shown to possess additional functions in mRNA metabolism[6–9] and tumour development[14,16]. Recent studies report that SRSF1 enhances cell mobility, invasion and drives oncogenic transformation of fibroblasts, and these features characterize SRSF1 as an oncogenic gene[14–16,34]. Importantly, SRSF1 has been shown to actively participate in the alternative splicing events that contribute to the phenotypic modulation between proliferating and differentiated VSMCs[17]. Here we provide direct evidence that SRSF1 signals play a critical role in the pathogenesis of neointima formation and VSMC proliferation. Particularly, we report that mice deficient in SRSF1 were protected from intimal hyperplasia in response to vascular injury. Reduced numbers of PCNA-positive VSMCs in injured arteries from $Srsf1^{-/-}$ mice and an inhibitory effect on apoptosis implied a pro-proliferative and pro-survival action of SRSF1 on VSMCs. Our present findings not only reveal a novel function of SRSF1 as an activator of VSMC proliferation but also have important implications for a role of SRSF1 in the development of atherosclerosis and perhaps other vascular proliferative disorders. For example, inhibition of SRSF1 might provide a potent and selective mechanism for reducing VSMC proliferation after injury.

We further found that Δ133p53, a shorter isoform of p53, is a downstream effector of SRSF1 in regulating VSMC proliferation. Δ133p53 is a human N-terminal-truncated p53 isoform that is initiated from an alternative p53 promoter in intron 4, lacking the transactivation domain and part of the DNA-binding domain of full-length p53 (ref. 35). It is a target of p53 and antagonizes p53-mediated apoptosis, represses cell replication senescence and promotes angiogenesis[35–37]. Here we showed that Δ133p53 strongly accumulated in the arteries that had been injured and after treatment with mitogenic factors, such as Ang II and PDGF-BB. Importantly, Δ133p53 was positively regulated by SRSF1, and only the Δ133p53 isoform, but not the other isoforms (full-length p53 and Δ40p53), was affected by SRSF1. SRSF1 has been reported to control several aspects of the cell cycle and growth via splicing-dependent actions, by which SRSF1 affects alternative splicing of the proto-oncogene RON[38] and the kinase S6K1 (ref. 15) to generate isoforms that promote cell motility, as well as splicing-independent pathways. In the splicing-independent pathways, SRSF1 interacts with and activates mammalian target of rapamycin[8] and phosphorylates the translation-activator S6 or elF4E and the translational-repressor 4EBP1 (ref. 15) to stimulate translation and cell proliferation. These studies have raised numerous possibilities for the way in which SRSF1 regulates Δ133p53 and the exact underlying mechanism merits further investigation.

In accord with the pro-proliferative effect of Δ133p53 reported previously[39,40], opposite to the inhibitory effect of p53, enforced expression of Δ133p53 accelerated HASMC proliferation and neointimal thickening, suggesting that, similar to SRSF1, Δ133p53 also acts as a positive regulator of VSMC proliferation.

Regarding the mechanisms responsible for the SRSF1- and Δ133p53-mediated activation of VSMC proliferation, we demonstrated that Δ133p53 interacts with EGR1, which in turn activates KLF5–p21 signalling. Using an RNA-seq approach, we first showed that KLF5, a member of the Krüppel-like transcription factor family and a crucial determinant of the cellular response to vascular injury[41,42], was enriched by introducing either Δ133p53 or Ang II into HASMCs. We revealed the involvement of KLF5 signalling in SRSF1-induced proliferation and that p21 is a downstream effector in terms of cell-cycle regulation[28]. Our results further showed that Δ133p53 formed a protein complex with EGR1, the early-response gene essential for the induction of various genes during vascular injury, and directly activates KLF5 transcription by binding the KLF5 promoter in VSMCs[30]. Through bioinformatics prediction[43] and CHIP-PCR, we verified that two EGR1-binding sites, characterized by GC-rich DNA motifs (such as GCGGGGGCG and CGCCCATGC)[30], in the KLF5 promoter region positively responded to Δ133p53 expression. These findings collectively establish a model in which SRSF1 upregulates Δ133p53, Δ133p53 interacts with EGR1 and facilitates EGR1 binding to the KLF5 promoter, which transcriptionally activates KLF5 and its target genes, resulting in improved cell-cycle progression, which, in turn, contributes to neointima formation and hyperplasia (Fig. 10n).

In concert with the pro-proliferative effect, SRSF1 and Δ133p53 facilitated VSMC migration. Moreover, they also promoted endothelial cell migration. In addition to regulating the cell cycle, induction of either EGR1 or KLF5 upregulates genes essential for migration and promotes the migration of various cells[44,45]. It is possible that activation of the EGR1–KLF5 signalling pathway also participates in SRSF1- and Δ133p53-induced cell migration. Δ133p53 is elevated in a number of human cancers[1,39] and was induced by injury in this study, suggesting an enriched expression pattern of Δ133p53 in proliferating cells. Moreover, Δ133p53 promotes proliferation and migration of multiple cell types, including VSMCs and endothelial cells in our study, and T cells, fibroblasts and cancer cells in previous reports[1,39,40]. Together with the proliferative and pro-migratory roles of EGR1 and KLF5 (refs 44,45), these findings suggest that Δ133p53–EGR1–KLF5 constitutes a 'normal' or 'general' signal following injury or during wound healing or perhaps other biological processes that involve migration and proliferation.

In addition, SRSF1 inhibits VSMC apoptosis, which leads to increased cell numbers and contributes to neointima generation[46] via upregulation of Bcl-xL protein. SRSF1 modulates global apoptotic splicing[47]. Specifically, SRSF1 regulates Bcl-x splicing, in which SRSF1 knockdown shifts the Bcl-x protein isoform toward Bcl-xS, the short and pro-apoptotic isoform, whereas SRSF1 overexpression favours antiapoptotic splicing, resulting in induction of Bcl-xL, the long and antiapoptotic isoform[47]. In our study, SRSF1 induced the expression of Bcl-xL, a well-established regulator crucial for VSMC apoptosis[48], highlighting the splicing regulation by SRSF1 in the vascular context. Moreover, this vascular SRSF1–Bcl-xL signal was also mediated by Δ133p53, which has been shown in zebrafish to antagonize p53-mediated apoptosis via regulating Bcl-xL[36], suggesting that Δ133p53 itself is a pro-survival factor as well.

In this study, we have demonstrated for the first time that upregulation of SRSF1 promotes, while its depletion attenuates VSMC proliferation and neointima formation in response to arterial injury or mitogenic stimuli. Because enhanced VSMC proliferation is essentially involved in the pathogenesis of many vascular proliferative disorders, antiproliferative therapy is beneficial in patients with such disorders, including atherosclerosis, in-stent coronary artery restenosis, the failure of

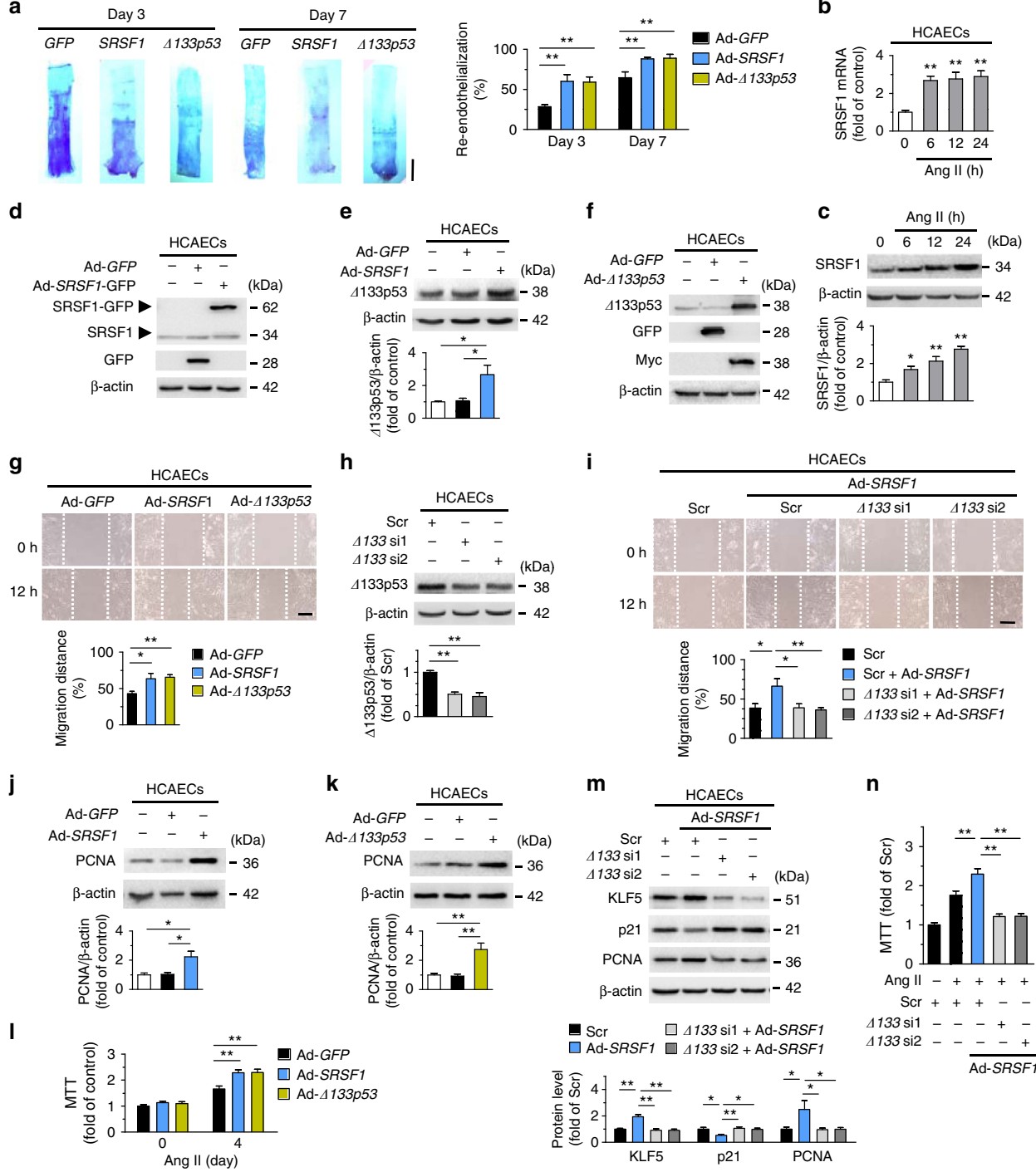

**Figure 9 | SRSF1 facilitates endothelia cell migration and proliferation.** (**a**) Representative pictures and averaged data of re-endothelialization. Re-endothelialization was quantified in Evans blue-stained carotid arteries at 3 and 7 days after vascular injury. Blue staining indicates endothelial denudation. $n = 9$ per group. Scale bar, 10 mm. (**b**) mRNA levels of SRSF1 in human coronary artery endothelia cells (HCAECs) treated with Ang II at 6, 12 and 24 h; $n = 8$ per group. (**c**) SRSF1 expression in HCAECs treated as in **a**; $n = 7$ per group. (**d**) Overexpression of SRSF1 by Ad-SRSF1 infection in HCAECs; $n = 5$ per group. (**e**) Δ133p53 levels in HCAECs infected with Ad-GFP or Ad-SRSF1; $n = 6$ per group. (**f**) Overexpression of Δ133p53 in cultured HCAECs by Ad-Δ133p53 infection; $n = 5$ per group. (**g**) Representative images and averaged data from wound-healing assays in HCAECs infected with Ad-GFP, Ad-SRSF1 or Ad-Δ133p53 and treated Ang II; $n = 9$ per group. Scale bar, 200 μm. (**h**) Δ133p53 siRNAs (Δ133 si1 and Δ133 si2) deplete Δ133p53 protein in HCAECs; $n = 6$ per group. (**i**) Representative images and averaged data from wound-healing assays in HCAECs infected with Δ133p53 siRNAs with or without Ad-SRSF1 overexpression and treated with Ang II; $n = 9$ per group. Scale bar, 200 μm. (**j,k**) PCNA expression in cultured HCAECs infected with Ad-SRSF1 (**j**) or Ad-Δ133p53 (**k**); $n = 5$ each. (**l**) MTT assays of HCAECs infected with Ad-GFP, Ad-SRSF1 or Ad-Δ133p53 after Ang II for 4 days; $n = 12$ per group. (**m**) KLF5, p21, and PCNA levels in HCAECs infected with scrambled or Δ133p53 siRNAs in the presence or absence of Ad-SRSF1 overexpression; $n = 7$ per group. (**n**) MTT assays showing the SRSF1-mediated enhancement of proliferation was inhibited by Δ133p53 knockdown in HCAECs; $n = 15$ per group. Adenoviral infection above is 100 m.o.i. for 48 h. Ang II concentration is 200 nM. Scr indicates scrambled siRNA control. *$P < 0.05$, **$P < 0.01$, one-way ANOVA (**a–c,g–n**). Data are mean ± s.e.m. of four independent experiments (**a–c,e,g–n**).

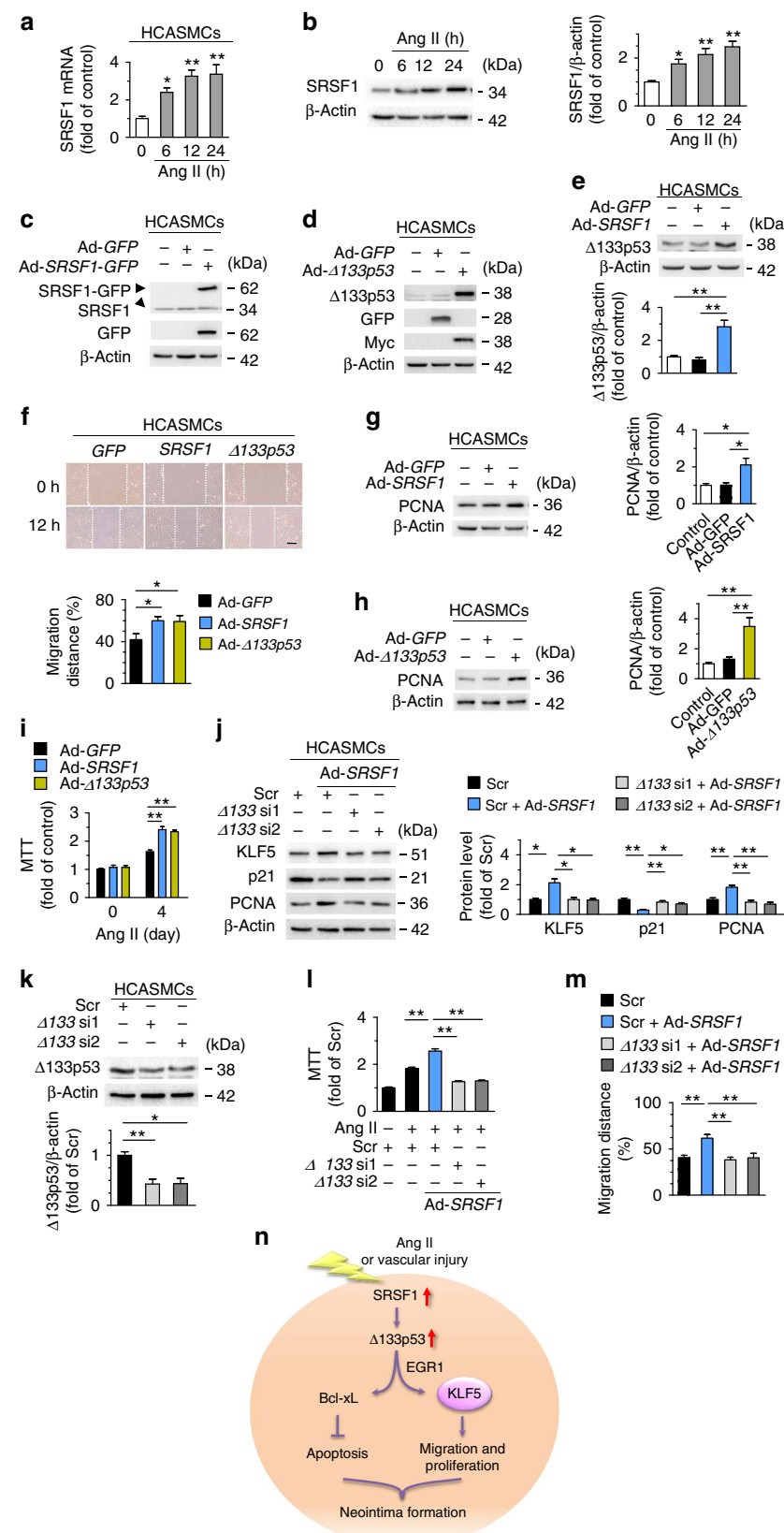

coronary arterial bypass grafts and narrowing of coronary arteries after cardiac transplantation[1,2]. In addition, SRSF1 deletion inhibits VSMC migration *in vivo* and *in vitro*. Since VSMC migration is another key feature of pathological vascular remodelling following injury[1,2], inhibiting cell migration is of therapeutic importance in preventing intimal hyperplasia. Thus

the antiproliferative and antimigratory effects of SRSF1 deletion could provide a potential novel therapy for the treatment of proliferative vascular diseases. What is more, SRSF1 deletion inhibits the growth and migration of both human aortic and coronary arterial SMCs, increasing the possibility of its utility as a therapeutic target in cardiovascular diseases. Taken together, our findings suggest that SRSF1 may serve as a novel therapeutic target for vascular disease. Meanwhile, with regard to the pro-migratory effect of SRSF1 on endothelial cells, anti-SRSF1 treatment might have thrombotic complications due to inadequate re-endothelialization. Thus treatment limiting endothelial cell migration or antithrombotic therapy should be included in potential translational studies targeting SRSF1.

In conclusion, our studies demonstrated a novel mechanism of VSMC migration, proliferation and neointimal hyperplasia. SRSF1 and SRSF1–Δ133p53–KLF5 signals play a major role in pathophysiological vascular hyperplasia and may have important therapeutic implications in restenosis, atherosclerosis and perhaps other human vascular proliferative diseases.

## Methods

**Animals.** All procedures involving experimental animals were performed using protocols approved by the Committee for Animal Research of Peking University, Beijing, China and in accordance with the Guide for the Care and Use of Laboratory Animals. All rats and mice were housed on a 12 h light, 12 h dark cycle, with free access to water and chow diet. The animals were randomly allocated to experimental groups. Two-to-3-month-old males were used for experiments. Male Sprague-Dawley rats used in carotid artery injury model were supplied by Vital River Laboratories, Beijing, China. $Srsf1^{flox/flox}$ mouse was generously provided by Dr X.D. Fu. $Sm22\alpha$-Cre and SMA-Cre$^{ERT2}$ mice were generously provided by Dr Z. Yang. No non-inclusion or exclusion parameters were used in our studies. Investigators were not blinded to treatments, but no subjective assessments were made.

**Human samples.** The human internal mammary arteries (Supplementary Fig. 2) were collected from 10 patients during cardiac bypass surgery. This study was approved by the Ethics Committee at Peking University Third Hospital, and written informed consent was given by all patients.

**Artery injury and morphometric analysis.** Balloon dilation of the left common carotid artery (LCCA) of male Sprague-Dawley rats (250–300 g) was performed as previously described[19,20]. Briefly, rats were anaesthetized by intraperitoneal injection of pentobarbital sodium (30 mg kg$^{-1}$ bodyweight). A 2 F embolectomy balloon catheter was inserted into the left common carotid via the external carotid. Then the inflated balloon was drawn gently towards the external carotid. After repeating this procedure three times, the catheter was removed. The injured artery was washed with phosphate-buffered saline and incubated for 20 min with 20 μl of adenoviral vector ($3 \times 10^9$ plaque-forming units ml$^{-1}$) expressing GFP, SRSF1, Δ133p53, p53 or Δ40p53. At the designated times, rats were killed with an overdose of pentobarbital sodium (100 mg kg$^{-1}$) by intraperitoneal injection and arteries were collected for western blot assays or embedded in paraffin to prepare cross-sections. Haematoxylin/eosin-stained arterial sections were analyzed by planimetry with ImageJ (http://imagej.nih.gov/ij/). The area of the residual lumen and the area circumscribed by the internal elastic lamina were measured directly. The degree of neointimal thickening was assessed using the intima-to-media area ratio and the neointimal area. The contralateral carotid arteries underwent the same procedures apart from injury and served as the sham group. Re-endothelialization of the carotid artery was determined by Evans blue staining 7 and 14 days after wire injury in rats.

Wire injury of the LCCA of male WT and $Srsf1^{-/-}$ mice (10–12 weeks of age) was performed as described by Lindner et al.[49]. In brief, the LCCA, including the bifurcation, was exposed and cleared from surrounding tissue. A 6-0 suture was placed around the left external carotid artery in which an incision and a flexible wire (0.38 mm) was introduced. The wire was removed, and the left external carotid artery was then tied off proximally. The arteries were harvested 28 days later and embedded in paraffin to prepare cross-sections. Haematoxylin/eosin-stained arterial sections were analyzed by planimetry with ImageJ (http://imagej.nih.gov/ij/). The contralateral carotid arteries underwent the entire procedure apart from injury and served as the sham group. Re-endothelialization of the arteries was determined by Evans blue staining 7 and 14 days after wire injury in mice. Similar procedures were performed on male WT and $Srsf1^{-/-}$ /Cre$^{ERT2}$ mice (10–12 weeks of age).

**Tamoxifen treatment.** Tamoxifen (Calbiochem) was prepared in corn oil at a concentration of 10 mg ml$^{-1}$. Male $Srsf1^{-/-}$ /Cre$^{ERT2}$ mice and control mice were injected intraperitoneally with 2 mg tamoxifen per day per 20 g body weight for 5 days to induce Cre-mediated deletion, and all studies were performed at 1 week after tamoxifen injection.

**Immunohistochemical and dual immunofluorescence assays.** The immuno-histochemical assays for PCNA were performed in sections from rat carotid arteries 14 days after injury and in sections from injured mouse carotid arteries 14 and 28 days after injury as previously described[19,20]. Briefly, endogenous peroxidase activity was quenched for 10 min in 3% H$_2$O$_2$. Then sections were incubated in blocking buffer (5% bovine serum albumin in PBS) for 1 h at room temperature, then incubated with mouse anti-PCNA antibody (Cell Signaling Tech, #2,586, 1:200 dilution) for 1 h and visualized with 3,3′-diaminobenzidine substrate (ZSGB-bio) followed by counterstaining with haematoxylin. The number of PCNA-positive nuclei and the total number of nuclei were counted in the neointima and media at ×200 magnification in a minimum of 250 cells per field. Six cross-sections from each artery were stained and counts were made in three fields from each section. The proportion of PCNA-positive cells was expressed as the ratio of the number of PCNA-positive nuclei to the total number of nuclei.

Dual immunofluorescence was performed in sections from rat carotid arteries 14 days after injury using the antibodies anti-SRSF1 (Abcam, ab129108, 1:200 dilution) and SM-α-actin (Abcam, ab5694, 1:200 dilution), and images were captured with a laser scanning confocal microscope (TCS-SP5, LEICA)[50]. The antibodies used are listed in Supplementary Table 2.

**Blood pressure and heart rate measurements.** Systolic and diastolic blood pressure and heart rate were measured non-invasively in conscious mice using a tail-cuff system warmed to 37 °C (Visitech BP-2000 Blood Pressure Analysis System). Mice were habituated to the device for 7–10 days and underwent two cycles of 10 measurements per day for 3 days.

**Recombinant adenovirus production.** The *SRSF1, Δ133p53, p53, Δ40p53* or *KLF5* gene was cloned and inserted into pEGFP-C1 vector (Clontech), then each gene was cloned and constructed into the pShuttle-CMV vector (Strategene). The adenovirus was produced using the AdEasy system (Strategene). Briefly, the recombinant plasmid pShuttle-CMV-srsf1, pShuttle-CMV-Δ133p53, pShuttle-CMV-p53, pShuttle-CMV-Δ40p53 or pShuttle-CMV-KLF5 was transformed into BJ5183-AD competent cells by electroporation to produce recombinant Ad plasmid. The adenovirus was produced in HEK293A cells by transfection of the recombinant Ad plasmid. Adenovirus containing the *GFP* gene alone served as control.

**Figure 10 | SRSF1 enhances human coronary artery SMC migration and proliferation.** (**a**) The mRNA levels of SRSF1 in human coronary artery smooth muscle cells (HCASMCs) treated with Ang II at 6, 12 and 24 h; $n = 8$ each. (**b**) SRSF1 levels in HCASMCs treated as in **a**; $n = 9$ each. (**c**) SRSF1 expression in cultured HCASMCs infected with Ad-*GFP* or Ad-*SRSF1*-GFP; $n = 5$ each. (**d**) Δ133p53 expression in cultured HCASMCs infected with Ad-*GFP* or Ad-*Δ133p53*; $n = 5$ each. (**e**) Δ133p53 levels in HCAECs infected with Ad-*GFP* or Ad-*SRSF1*; $n = 6$ each. (**f**) Representative images and averaged data from wound-healing assays in HCASMCs infected with Ad-*GFP*, Ad-*SRSF1* or Ad-*Δ133p53* and treated with Ang II; $n = 9$ each. Scale bar, 200 μm. (**g,h**) PCNA expression in cultured HCAECs infected with Ad-*SRSF1* (**g**) or Ad-*Δ133p53* (**h**); $n = 5$ each. (**i**) MTT assays of HCASMCs infected with Ad-*GFP*, Ad-*SRSF1* or Ad-*Δ133p53* (m.o.i. 100, 48 h) after Ang II for 4 days; $n = 12$ each. (**j**) KLF5, p21 and PCNA levels in HCASMCs infected with *Δ133p53* siRNAs in the presence or absence of Ad-*SRSF1* overexpression; $n = 7$ each. (**k**) Δ133p53 levels in HCASMCs infected with *Δ133p53* siRNAs (*Δ133* si1 and *Δ133* si2); $n = 7$ each. (**l**) MTT assays showing the SRSF1-mediated enhancement of proliferation was inhibited by Δ133p53 knockdown in HCASMCs; $n = 15$ each. (**m**) Averaged data from wound-healing assays in HCASMCs infected with *Δ133p53* siRNAs with or without Ad-*SRSF1* overexpression and treated with Ang II; $n = 5$ each. (**n**) Schematic outline of SRSF1 -D133p53 -KLF5 axis in VSMC. Ang II, angiotensin II; SRSF1, serine/arginine-rich splicing factor 1; EGR1, early-growth-response gene 1; KLF5, Krû¥ppel-like factor 5; VSMC, vascular smooth muscle cell. Scr indicates scrambled siRNA control. All adenoviral infection above is 100 m.o.i. for 48 h. Concentration of Ang II is 200 nM. Scr indicates scrambled siRNA control. *$P < 0.05$, **$P < 0.01$, one-way ANOVA (**a,b,e-m**). Data are mean ± s.e.m. of four independent experiments (**a,b,e-m**).

**Cell culture and adenoviral infection.** HASMCs were obtained from ScienCell Research Laboratories (Catalog #6,110), and HCASMCs and HCAECs were from PromoCell Research Laboratories (Catalog #C-12511, #C-12221); all the cells were P0 or P1 primary cells, not immortal cell lines. HCASMCs and HASMCs were cultured in SMC medium (ScienCell, 1101) while HCAECs were cultured in endothelial cell medium (ScienCell, 1001); they were used at passages 4–7. HASMC, HCASMC and HCAEC synchronization were achieved by starving the cells in SMC medium or endothelial cell medium for 24 h followed by Ad-GFP, Ad-SRSF1, Ad-Δ133p53, Ad-p53, Ad-Δ40p53 or Ad-KLF5 (Krüppel-like factor 5) at a multiplicity of infection of 50 or 100 (50 or 100 plaque-forming units per cell) for 48 h. The infection efficiency was assessed by western blot for each adenoviral vector.

**MTT assay and cell counts.** Mitogenic quiescence was induced in HASMCs seeded in 24-well plates at $5 \times 10^3$ cells per well by serum starvation for 24 h. The cells were then infected with adenovirus for 48 h followed by stimulation with Ang II (200 nM), serum (10% FBS) or PDGF-BB (10 μg l$^{-1}$). The growth curves of VSMCs were constructed from cell counts on the indicated days after infection. Cell proliferation was also assayed by cleavage of the tetrazolium salt 3-(4,5-dimethylthiazol-2-yl)-2,5-diphenyltetrazolium bromide (MTT). After cells were incubated with MTT for 4 h at 37 °C, the medium was removed, and the cells were solubilized in 450 μl dimethylsulfoxide. The MTT cleavage was quantified spectrophotometrically at 490 nm with background subtraction at 630 nm[19,20].

**Cell migration assay.** HASMCs, HCASMCs or HAECs were infected with Ad-SRSF1, Ad-Δ133p53, Ad-p53, Ad-Δ40p53 or Ad-GFP for 48 h before scratch assay. For the scratch assay, cells ($3 \times 10^5$) were seeded in six-well plates, rendered quiescent by serum starvation for 24 h and then plated in a six-well plate. The scratch was made and fresh medium that contained 10% FBS was added. Images were captured at 0 and 12 h after treatment.

**Western blot analysis.** Proteins were prepared from HASMCs or arteries. For western blot analysis, proteins were electrophoresed on 10% SDS–polyacrylamide gel electrophoresis and transferred to a polyvinylidene difluoride membrane (Bio-Rad). Each membrane was blocked with 5% nonfat dry milk in Tris-buffered saline and 0.1% Tween 20 (TBS-T) for 1 h and then incubated with the primary antibody at 4 °C overnight. The membrane was washed and incubated with the secondary antibody conjugated with horseradish peroxidase in 5% non-fat milk in TBS-T buffer for 1 h. Detection was carried out using a Chemiluminescence Detection Kit (Cell Signaling Technology)[50,51]. Uncropped images of blots are presented in Supplementary Fig. 20.

**Co-immunoprecipitation assay.** Cells were lysed in lysis buffer A (30 mM Hepes (pH 7.6), 100 mM NaCl, 0.5% Nonidet P-40 and protease inhibitor mixture) on ice for 10 min, and the lysates were clarified by centrifugation at 4 °C for 10 min at 13,000 r.p.m. The supernatant was mixed with nProtein A Sepharose 4 Fast Flow (GE Healthcare) and the antibody and incubated at 4 °C for 2 h, using antibody anti-Myc (1:100 for immunoprecipitation, 1:1,000 for western blots) or anti-EGR1 (1:100 for immunoprecipitation, 1:1,000 for western blots). The resins were then washed three times with lysis buffer A, and the bound proteins were detected by western blotting[51]. The antibodies used are listed in Supplementary Table 2.

**Real-time PCR.** RNA was extracted from cells or tissues using TRIzol reagent (Sigma). Following cDNA synthesis (M-MLV reverse transcriptase, Promega), the relative levels of mRNA were evaluated by real-time PCR (SteponePlus Real-Time PCR System, Applied Biosystems). The relative mRNA levels were determined by normalization to the 18S RNA level[19,20,50,51]. The primers used are listed in Supplementary Table 3 and Supplementary Tables 6 and 7. Data presented are the average of at least four independent experiments.

**Small-interfering RNAs.** Small-interfering RNA (siRNA) duplexes for silencing SRSF1, EGR1, KLF5 and p21 were designed using RNAi Designer and synthesized by Genepharma. Both Δ133p53 siRNA1 and Δ133p53 siRNA2 were designed to target the sequences present in Δ133p53 mRNA as the 5′ untranslated region but spliced out of full-length p53 mRNA as intron 4 (ref. 52).

Transfection of siRNAs was performed with Lipofectamine RNAiMax (Invitrogen) according to the manufacturer's protocol. HASMCs were seeded in 60-mm dishes at $5 \times 10^5$ cells per dish and then infected with siRNA. siRNA duplexes for silencing SRSF1, Δ133p53, KLF5 and EGR1 were designed using RNAi Designer and synthesized by Genepharma[50,51]. The sequences of the siRNAs against SRSF1, Δ133p53, KLF5 and EGR1 are listed in Supplementary Table 4.

**RNA-Seq.** Total RNA was extracted from HASMCs with Δ133p53 overexpression, GFP overexpression or Ang II treatment (200 nM). PolyA+ RNAs were purified and sequenced on an Illumina HiSeq2500 system according to the manufacturer's instructions. For each condition, RNA-seq libraries were prepared and sequenced from three samples of HASMCs from each group. RNA-Seq reads were mapped to the reference human genome (hg19) by TopHat2 (v2.0.12)[18]. HTSeq (v0.6.1, union mode) was used to count the number of reads aligned to each gene and differential expression analysis was performed using the DESeq package (v1.18.0)[26,53]. All sequencing data have been submitted to the NCBI Sequence Read Archive, under accession number SRP090170.

**Chromatin immunoprecipitation assay.** The ChIP assay was performed with a kit from Cell Signaling Technology (catalogue number 9003) followed by quantitative real-time PCR. The primers used for ChIP-PCR are listed in Supplementary Table 5.

**Flow cytometry.** Cell-cycle detection was carried out by propidium iodide staining and fluorescence-activated cell sorter analysis. HASMCs were synchronized, infected with adenoviral vectors for 48 h and then stimulated to proliferate with Ang II as described above. They were fixed overnight in 75% ethanol at 4 °C and stained for 10 min at room temperature with propidium iodide (50 μg ml$^{-1}$). Cells were analyzed with a FACSCanto flow cytometer (BD Biosciences). A total of $10^4$ cells were counted for each sample, and the Double Discriminator Module was used to detect single cells. Each experiment was repeated three times.

**Cell viability assays.** VSMC viability was assessed by an ATP assay as previously described[54]. The CellTiter-Glo Luminescent Cell Viability Assay Kit (Promega, catalogue number G7570) was used for ATP assay and luminometer (Bio-Tek, USA) for the luminometric measurement. The count is directly proportional to the cellular ATP content.

DNA laddering was performed as previously described[50]. Briefly, cells were lysed in lysis buffer (150 mM NaCl, 10 mM Tris-Cl (pH 8.0), 0.4% SDS, 10 mM EDTA and 100 μg ml$^{-1}$ protease K) after adenoviral infection or siRNA transfection; incubated at 50 °C for 5 h and then extracted with phenol/CHCl3/ isoamyl alcohol, followed by CHCl$_3$/isoamyl alcohol. To detect DNA fragmentation, 10 μg of total DNA was loaded onto 2% agarose gel in Tris acetate/EDTA buffer and visualized by ethidium bromide staining.

**TUNEL staining.** The CardioTACS In Situ Apoptosis Detection Kit (Roche Applied Science, catalogue number 11684795910) was used for TUNEL staining, which was performed in sections from rat carotid arteries 14 days after injury and in sections from mouse carotid arteries 28 days after injury[55].

**Caspase 3 activity assay.** Caspase 3 activity was measured with a kit from Promega (G8091)[55].

**Materials.** Antibodies directed against the following proteins are listed in Supplementary Table 2: SRSF1, SM-α-actin, PCNA, p21, EGR1, GFP, Myc, p53, β-actin, Eif-5, KLF5, Bcl-xL.Bcl-xL Angiotensin II (Ang II), and PDGF-BB were from Sigma. Unless indicated otherwise, all other chemicals were from Sigma-Aldrich.

**Statistical analysis.** All data are expressed as the mean ± s.e.m. Statistical analyses were performed with GraphPad PRISM version 5.01 (GraphPad Software, Inc.) and SPSS 20.0 (SPSS Inc.). Data sets were tested for normality of distribution with Kolmogorov–Smirnov tests. Data groups (two groups) with normal distributions were compared using two-sided unpaired Student's t-tests. The Mann–Whitney U-test was used for nonparametric data. Comparisons among groups were analyzed by one-way analysis of variance with Bonferroni or Dunnett's post hoc analysis. For the cell growth experiments, repeated-measures analysis of variance was used to compare cell growth in control and treated HASMCs over the time course. $*P < 0.05$; $**P < 0.01$; NS, not significant. No statistical method was used to predetermine sample size.

**Data availability.** All sequencing data have been submitted to the NCBI Sequence Read Archive under accession number SRP090170. The authors declare that all other data supporting the findings of this study are available within the article and its files.

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

## Acknowledgements

We thank X.D. Fu for the floxed SRSF1 mice and Z. Yang for the SM22α-Cre and SMA-Cre[ERT2] mice; L.P. Wei and I.C. Bruce for comments on the manuscript and L. Jin, X. Cui, P. Xiao, and J. Fu for excellent technical support. This work was supported by the National Natural Science Foundation of China (91439107, 81170100 and 81370233 to C.-M.C. and 81370234 to Y.Z.), CAMS Initiatives for Innovative Medicine (CAMS-12M) (to C.-M.C.), a National Science and Technology Support Project (2014BAI02B01 to C.-M.C.) and a Beijing Talents Project (to C.-M.C.).

## Author contributions

N.X. and M.C. generated the initial idea and conducted key experiments. R.D., Y.Z., H.Z., Z.S., L.Z., Z.L., Y.F., H.G., L.W., T.Z., R.-P.X. and J.W. performed experiments. N.X. and C.-M.C. proposed the hypothesis, designed the study, supervised the experiments and wrote and edited the manuscript.

## Additional information

**Competing interests:** The authors declare no competing financial interests.

