## [Peer Review File · Nature Communications]

Reviewers' comments:

Reviewer #1 (expert in p53 biology)

Remarks to the Author:

Review of Xie et al N Comm

The authors investigate a role for the splicing factor SRSF1 in promoting proliferation in vascular smooth muscle cells (VSMCs). Proliferation and migration of these cells occur in response to injury resulting in expansion of the arterial intima wall which and can cause hypertension and worse, perhaps, resulting in a heart attack or stroke. Thus investigating the mechanism of cell wall expansion after injury is an important area of investigation.

Overall this is an interesting paper for both the cardiac pathology field and also it provides (in principle – see comments below) an important physiological function for the D133p53 in promoting muscle proliferation in response to injury. This finding is consistent with a recent report showing that D133p53 plays a key migratory role in wound healing experiments (PMID: 26996665); which should be cited.

Mainly the studies have been well done, are thorough and generally support the authors' conclusions. It is also fairly well written.

I do have some concerns which are identified below:

Results:

They show in Fig 1 that SRSF1 is expressed in cardiac vessels which increases upon injury and other mitogenic stimulation. This is followed up by creating an SRSF1 KO mouse in aortic cells and confirming that the KO cells are impaired for expansion/proliferation of arterial tissue in response to injury. That SRSF1 promotes proliferation is confirmed in Fig 3 by over-expressing SRSF1 in human smooth muscle cells. These data mostly support the conclusions.

Critique: the details of the SRSF1 and PCNA antibodies should be provided as well as some evidence of specificity. This could include in Supplementary material, a western blot showing a single protein species, full length gels, combined with siRNAs to the targets on the same gel.

One question: It is not clear why SRSF1 was chosen as the focus of the project. The introduction just provides evidence that SRSF1 promotes proliferation, not that it has any particular role in VSMCs. Some discussion of that point would be useful, otherwise it just appears to be a random choice as many genes satisfy the condition of being promoters of proliferation.

In a similar vein, they discovered a "serendipitous" link between SRSF1 and the D133p53 isoform. How?

In Fig 4, they show that D133p53 mRNA protein are up-regulated by over-expression of SRSF1 and in Supplementary Fig 4 they show that neither the full length (FL) p53 mRNA nor the D40p53 isoform mRNAs change, concluding that the effect of SRSF1 is specific to the D133p53 isoform. But p53 is mostly regulated post-transcriptionally so the authors should also show western blots for these other p53 species. In addition, the authors should provide the antibody details, including the location of the epitopes – especially what antibody was used to detect the mouse D157p53 isoform. Given that there are no isoform specific antibodies, the D133 (and D157) species can only be determined by size on

gels. Therefore, full gels should be provided as Supplementary material; and for all the p53 species.

Also, do the authors know if the D133p53 they are studying is the alpha, beta or gamma variants?

That said, the biological effects of the (putative) D133p53 in Fig 4 do seem to be clear and recapitulate the phenotype of SRSF1. A useful control however would be to do similar experiments with FLp53 and the D40 species.

The authors go on in Fig 5 to show that D133p53 and SRSF1 transcriptionally regulate the KLF5 transcription factor which is the key cell cycle regulator in human aortic muscle cells. These data seem convincing as do some rather tidy ChIP experiments (Fig 6) showings that D133p53 forms a complex with Early Response Gene (EGR)-1 which binds to the KLF5 promoter thus upregulating expression of KLF5. In Fig 7 they also show that D133p53 blocks apoptosis by upregulating bcl2; hardly a surprising finding which could probably be relegated to supplementary material.

Discussion:

Much of this is a summary of the results and so could be considerably shortened and more attention paid to the physiological implications of the D133/egr1/KLF5 axis in aortic muscle injury and perhaps in other "injury/pathology" contexts.

References:

The papers cited for D133p53 promoting proliferation I think are incorrect – the Fujita paper shows is mainly about p53beta promoting senescence and the Chen paper shows that zebrafish homologue protects from cell death – there is little about proliferation.

The best reference for D133p53 promoting proliferation is PMID:24231352 but the first reference was PMID:26996665.

Reviewer #2 (expert in vascular smooth muscle cells and neointima formation)

Remarks to the Author:

In this manuscript, Xie et al describe that "SRSF1 promotes vascular smooth muscle cell proliferation through a delta133p53/KLF5 pathway."

In general, this laborious study demonstrates for the first time a functional role of SRSF1 in smooth muscle cells. However, there are some concerns which largely limit the scientific quality as well as the conclusions of this study.

Major:

- 1.) The expression of SRSF1 was upregulated in injured rat carotid arteries together with the proliferation marker PCNA. However, neither expression of SRSF nor neointima formation was assessed at later time points when neointima formation is completed and VSMC return to a non-proliferating phenotype. These data should be added and SRSF1 expression as well as definitive extent of neointima formation should be assessed after completion of the proliferative response (i.e. after 21 or 28 days) to exclude that neointima formation is really reduced, not only retarded.
- 2.) In general the reviewer favours the idea to complement in vivo data from animal models with in vitro experiments performed in human cells to strengthen a translational approach. However, it is not clear why the authors used human aortic smooth muscle cells. VSMC from different vascular beds differ largely in their behaviour and since injury-induced neointima formation occurs in smaller arteries like coronary arteries or peripheral arteries after angioplasty, the in vitro experiments in human VSMC

only make sense when they are performed in the adequate VSMC, i.e. human coronary artery smooth muscle cells which can easily be used instead of aortic VSMC. Thus, the key in vitro experiments should be repeated in human coronary artery smooth muscle cells.

Otherwise there might be the concern that the SM22a-Cre mice do not express Cre in VSMC appropriately so that a more specific Myh11-Cre mouse strain should be used.

3.) The authors represent laborious work demonstrating a loose upstream/downstream interconnection of the delta133p53/KLF5 pathway. However, they reproduce more or less a bunch of previous studies, which described the respective molecular interactions in more detail. Thus an additional gain of knowledge regarding the molecular action and the regulation of the delta133p53/KLF5 pathway is lacking.

4.) In (SRSF1flox/flox · SM22 (-Cre) mice, the SRSF1 expression was found to be only partially downregulated and the authors claim that this is due to the presence of different cell types in the aorta. To rule out this option, the media of these mice should be isolated and the SRSF1 expression should be determined to confirm that SRSF1 expression is indeed knocked out in the media, which comprises of more than 95% of VSMC in uninjured arteries.

5.) Does SMC-specific deletion of SRSF1 have any effect on the size of the media following injury (add data to Fig. 2C).

6.) Fig. 7a+g: neointimal lesions should be co-stained with TUNEL and a specific VSMC marker to confirm that apoptotic cells are indeed VSMC.

7.) Since it was reported that delta133p53 promotes angiogenesis it is likely that a knock down or knock out will also affect endothelial cell function during re-endothelialization. This in fact may have a profound impact on neointima formation. In this regard the authors should also assess the re-endothelialization following vascular injury in this model. Inhibition or knock down/knock out of SRSF1 might result in incomplete re-endothelialization and vessel healing despite a reduced neointimal lesion size. Such a lesion might be more thrombogenic and would prevent such a translational approach from being followed up in further pre-clinical and clinical studies. Thus the effects of SRSF1 on endothelial function and recovery in vitro and in vivo have to be included in the study.

8.) PCNA staining to determine the number of proliferating cells after injury of the mouse artery were performed at 28 days after injury. However, this is a time point where usually vascular remodelling and VSMC proliferation is completed in this model. Therefore it is quite astonishing that proliferative VSMC can be found. Experiments to determine VSMC proliferation in cross sections of injured mouse arteries should be repeated at 10-14 days after injury.

9.) The manuscript suffers from a general lack of correct grammar and language and should be extensively revised by a native speaker.

10.) Migration of VSMC is a further key feature of pathologic vascular remodelling following injury. Thus, the authors should determine the effects of SRSF1-induced delta133p53 expression on VSMC migration.

Minor points:

p.1 l.1: correct "KLF5 pathway..." to "KLF5 pathway"

p.2 l.72: "...expression is increased in proliferating VSMC..."

p.5 l.121: "...overexpression was almost double that in the Ad-GFP..."

p.6 l.143: "In mice, an isoform similar to human δ 133p53 is highly homologous, it was previously reported..."

p.14 l.327: "...and contributes to the generation of neointima ..."

p.18 l.422: in the methods the authors indicate that "None of the cell lines used were reported in the ICLAC database of commonly misidentified cell lines. All cell lines were tested monthly for mycoplasma contamination and found negative. After their initial purchase, cell lines were not further authenticated."

However, it is unclear what cell lines are meant since the authors describe that only primary cells were used for all experiments?

Point-by-point Reply

Reviewer #1 (expert in p53 biology)

The authors investigate a role for the splicing factor SRSF1 in promoting proliferation in vascular smooth muscle cells (VSMCs). Proliferation and migration of these cells occur in response to injury resulting in expansion of the arterial intima wall which and can cause hypertension and worse, perhaps, resulting in a heart attack or stroke. Thus investigating the mechanism of cell wall expansion after injury is an important area of investigation.

Overall this is an interesting paper for both the cardiac pathology field and also it provides (in principle – see comments below) an important physiological function for the D133p53 in promoting muscle proliferation in response to injury. This finding is consistent with a recent report showing that D133p53 plays a key migratory role in wound healing experiments (PMID: 26996665); which should be cited.

Per your suggestion, we have cited this study in the revised manuscript (revised References #38).

Mainly the studies have been well done, are thorough and generally support the authors' conclusions. It is also fairly well written.

I do have some concerns which are identified below:

Results:

They show in Fig 1 that SRSF1 is expressed in cardiac vessels which increases upon injury and other mitogenic stimulation. This is followed up by creating an SRSF1 KO mouse in aortic cells and confirming that the KO cells are impaired for expansion/proliferation of arterial tissue in response to injury. That SRSF1 promotes proliferation is confirmed in Fig 3 by over-expressing SRSF1 in

human smooth muscle cells. These data mostly support the conclusions.

Critique: the details of the SRSF1 and PCNA antibodies should be provided as well as some evidence of specificity. This could include in Supplementary material, a western blot showing a single protein species, full length gels, combined with siRNAs to the targets on the same gel.

Your point is well taken. We have now provided the antibody details and performed additional experiments to show the specificity of the SRSF1 and PCNA antibodies (revised **Supplementary Figure 18**). To determine the antibody specificity, we transfected human arterial smooth muscle cells (HASMCs) with SRSF1 siRNAs or PCNA siRNAs and detected the SRSF1 or PCNA protein by western blot. The SRSF1 antibody, mouse anti-SF2/ASF mAb (Invitrogen, Cat No. 32-4500), recognizes the first 97 amino-acids of SRSF1 (Dong and Chen, 2009). The PCNA antibody, PCNA(PC10) mouse mAb (Cell Signaling Technology, #2586), recognizes the protein region at amino-acid residues 111–125 (Roos et al., 1993). Single bands in the full gel showed that SRSF1 was effectively knocked down by siRNA transfection, suggesting high specificity of the SRSF1 antibody (revised **Supplementary Figure 18a**). The western blot for PCNA knockdown also showed good specificity of the PCNA antibody (revised **Supplementary Figure 18b**).

One question: It is not clear why SRSF1 was chosen as the focus of the project. The introduction just provides evidence that SRSF1 promotes proliferation, not that it has any particular role in VSMCs. Some discussion of that point would be useful, otherwise it just appears to be a random choice as many genes satisfy the condition of being promoters of proliferation.

In a similar vein, they discovered a “serendipitous” link between SRSF1 and the D133p53 isoform.

How?

Thanks for your important suggestions. The active participation of SRSF1 in alternative splicing events in shaping the transcriptome of proliferative VSMCs (Llorian et al., 2016) inspired us to study the function of SRSF1 in VSMCs. We have revised the introduction and discussion to raise this point. Our discovery of the regulatory action between SRSF1 and $\Delta 133p53$ initiated from our examination of SRSF1 with the well-studied cell growth regulator p53. In the revised manuscript, this is now presented.

In Fig 4, they show that $\Delta 133p53$ mRNA protein are up-regulated by over-expression of SRSF1 and in Supplementary Fig 4 they show that neither the full length (FL) p53 mRNA nor the $\Delta 40p53$ isoform mRNAs change, concluding that the effect of SRSF1 is specific to the $\Delta 133p53$ isoform. But p53 is mostly regulated post-transcriptionally so the authors should also show western blots for these other p53 species. In addition, the authors should provide the antibody details, including the location of the epitopes – especially what antibody was used to detect the mouse $\Delta 157p53$ isoform. Given that there are no isoform specific antibodies, the $\Delta 133$ (and $\Delta 157$) species can only be determined by size on gels. Therefore, full gels should be provided as Supplementary material; and for all the p53 species.

Your points are well taken. The antibody we used to detect $\Delta 133p53$ (and $\Delta 157p53$) was p53(CM1) antibody (Vector Laboratories, VP-P955), a rabbit polyclonal antibody raised against recombinant full-length human p53 protein that recognizes all p53 isoforms, in which $\Delta 133p53\beta$ and $\Delta 133p53\gamma$ are weakly recognized as these isoforms have lost most of the immunogenic domains (Bourdon, 2007; Midgley et al., 1992). The p53(CM1) antibody has been widely used to detect the $\Delta 133p53$ isoform or $\Delta 133p53$ homologue (Fujita et al., 2009; Murray-Zmijewski et al., 2006; Wei et al., 2012)

and its details are presented in the revised manuscript. The full gels are shown in revised **Supplementary Figure 19** for all p53 species.

Also, do the authors know if the $\Delta 133p53$ they are studying is the alpha, beta or gamma variants?

The $\Delta 133p53$ we studied was $\Delta 133p53\alpha$, the alpha variant of $\Delta 133p53$. To determine the variants of $\Delta 133p53$ in HASMCs, we used specific primers to amplify the α , β , and γ isoforms by nested RT-PCR, with human breast adenocarcinoma cells (MCF-7) as a positive control (Khoury et al., 2013). Then DNA sequencing was performed on the PCR products to verify the isoforms. Our data showed that while all three were present in MCF-7 cells, only $\Delta 133p53\alpha$ was detectable in HASMCs, suggesting that the $\Delta 133p53$ (at least its predominant form) present in HASMCs is $\Delta 133p53\alpha$ (revised **Supplementary Figure 5**). Besides, the recombinant adenovirus we used to express $\Delta 133p53$ was constructed by inserting the human $\Delta 133p53\alpha$ coding sequence, so that we overexpressed and studied the role of the alpha variant of $\Delta 133p53$ in vascular smooth muscle cells.

*That said, the biological effects of the (putative) $\Delta 133p53$ in Fig 4 do seem to be clear and recapitulate the phenotype of *SRSF1*. A useful control however would be to do similar experiments with *FLp53* and the *D40* species.*

Per your suggestions, we have performed additional experiments with *FLp53* and $\Delta 40p53$ as controls (see revised **Figure 4**). We overexpressed *FLp53* and $\Delta 40p53$ via adenovirus gene transfer to determine their effects on VSMC proliferation, migration, and neointima formation, as controls for $\Delta 133p53$ (revised Figure 4). Contrary to Ad- $\Delta 133p53$, both Ad-*FLp53* and Ad- $\Delta 40p53$ inhibited VSMC migration and proliferation *in vitro* and ameliorated neointima formation *in vivo* (revised

Fig. 4g-m), which is consistent with the inhibitory role of FLp53 in vascular remodeling previously reported (Yonemitsu et al., 1998).

The authors go on in Fig 5 to show that D133p53 and SRSF1 transcriptionally regulate the KLF5 transcription factor which is the key cell cycle regulator in human aortic muscle cells. These data seem convincing as do some rather tidy ChIP experiments (Fig 6) showings that D133p53 forms a complex with Early Response Gene (EGR)-1 which binds to the KLF5 promoter thus upregulating expression of KLF5. In Fig 7 they also show that D133p53 blocks apoptosis by upregulating bcl2; hardly a surprising finding which could probably be relegated to supplementary material.

Per your suggestion, we have moved **Fig. 7** to supplementary material (revised **Supplementary Figure 17**).

Discussion:

Much of this is a summary of the results and so could be considerably shortened and more attention paid to the physiological implications of the D133/egr1/KLF5 axis in aortic muscle injury and perhaps in other “injury/pathology” contexts.

Thank you for the suggestion. We have revised the discussion with a shortened summary of results and more attention focused on the physiological and pathological implications of our study.

References:

The papers cited for D133p53 promoting proliferation I think are incorrect – the Fujita paper shows is mainly about p53beta promoting senescence and the Chen paper shows that zebrafish

homologue protects from cell death – there is little about proliferation.

The best reference for D133p53 promoting proliferation is PMID:24231352 but the first reference was PMID:26996665.

We have updated the references in the revised manuscript per your thoughtful suggestion.

Reviewer #2 (expert in vascular smooth muscle cells and neointima formation)

Remarks to the Author:

In this manuscript, Xie et al describe that “SRSF1 promotes vascular smooth muscle cell proliferation through a delta133p53/KLF5 pathway.” In general, this laborious study demonstrates for the first time a functional role of SRSF1 in smooth muscle cells. However, there are some concerns which largely limit the scientific quality as well as the conclusions of this study.

Major:

1.) The expression of SRSF1 was upregulated in injured rat carotid arteries together with the proliferation marker PCNA. However, neither expression of SRSF nor neointima formation was assessed at later time points when neointima formation is completed and VSMC return to a non-proliferating phenotype. These data should be added and SRSF1 expression as well as definitive extent of neointima formation should be assessed after completion of the proliferative response (i.e. after 21 or 28 days) to exclude that neointima formation is really reduced, not only retarded.

Per your thoughtful suggestions, we have conducted additional experiments to determine the expression of SRSF1 and PCNA at later time points (21 days post-injury). Compared to that at 14

days, the immunohistochemical staining in injured arteries at 21 days showed a reduced number of SRSF1-positive cells, and a decrease in SRSF1 at the mRNA and protein levels, together with a decrease of PCNA at 21 days (revised **Fig. 1a,c,d**). These data, together with our previous results that SRSF1 was upregulated at 4, 7, and 14 days post-injury, suggest that SRSF1 expression is more abundant in proliferative than in non-proliferative VSMCs.

2.) In general the reviewer favours the idea to complement in vivo data from animal models with in vitro experiments performed in human cells to strengthen a translational approach. However, it is not clear why the authors used human aortic smooth muscle cells. VSMC from different vascular beds differ largely in their behaviour and since injury-induced neointima formation occurs in smaller arteries like coronary arteries or peripheral arteries after angioplasty, the in vitro experiments in human VSMC only make sense when they are performed in the adequate VSMC, i.e. human coronary artery smooth muscle cells which can easily be used instead of aortic VSMC. Thus, the key in vitro experiments should be repeated in human coronary artery smooth muscle cells.

Thank you for the thoughtful comments. We have performed three key experiments in cultured human coronary artery smooth muscle cells (HCASMCs). Our results showed that the expression of SRSF1 increased in HCASMCs in response to Ang II stimulation. Overexpression of SRSF1 enhanced the migration and proliferation of HCASMCs and up-regulated the $\Delta 133p53/KLF5$ signals, as is the case with human aortic SMCs (revised **Fig. 9**).

Otherwise there might be the concern that the SM22a-Cre mice do not express Cre in VSMC appropriately so that a more specific Myh11-Cre mouse strain should be used.

To thoroughly address your concern, we used another strain of inducible SMA-Cre^{ERT2} mice to confirm our results with SM22 α -Cre, since we had SMA-Cre^{ERT2} mice in-house. We obtained inducible smooth muscle cell-specific SRSF1-KO mice (*Srsf1*^{-/-}/Cre^{ERT2}) by crossing floxed SRSF1 mice with SMA-Cre^{ERT2} transgenic mice (SRSF1^{flox/flox} \times SMA-Cre^{ERT2}) (genotyping PCR in revised **Fig. 3a**). Western blots showed that SRSF1 was fully knocked down in the arterial media from *Srsf1*^{-/-}/Cre^{ERT2} mice, but not in the liver (revised **Fig. 3b**), indicating the specific deletion of SRSF1 in VSMCs. Tamoxifen-induced deletion of SRSF1 in *Srsf1*^{-/-}/Cre^{ERT2} mice attenuated the wire injury-induced neointima formation at 14 and 28 days after injury, and reduced the number of PCNA-positive cells, compared with WT littermate controls (revised **Fig. 3c,d**). These data further support the hypothesis that SRSF1-deficiency suppresses neointima formation.

3.) The authors represent laborious work demonstrating a loose upstream/downstream interconnection of the delta133p53/KLF5 pathway. However, they reproduce more or less a bunch of previous studies, which described the respective molecular interactions in more detail. Thus an additional gain of knowledge regarding the molecular action and the regulation of the delta133p53/KLF5 pathway is lacking.

We agree with you that uncovering the molecular mechanism underlying the regulation of the Δ 133p53/KLF5 pathway is critically important. At present, the mechanism is unclear. This, indeed, merits future investigation.

4.) In (SRSF1^{flox/flox} · SM22^{-/-} (-Cre)) mice, the SRSF1 expression was found to be only partially downregulated and the authors claim that this is due to the presence of different cells types in the aorta. To rule out this option, the media of these mice should be isolated and the SRSF1 expression

should be determined to confirm that SRSF1 expression is indeed knocked out in the media, which comprises of more than 95% of VSMC in uninjured arteries.

Your point is well taken. We have now rigorously examined the SRSF1 expression profile in *Srsf1*^{-/-} mice (SRSF1^{fllox/fllox}; SM22 α -Cre⁺) mice and confirmed that SRSF1 was completely deleted in the isolated media of these mice (revised **Fig. 2b**).

5.) Does SMC-specific deletion of SRSF1 have any effect on the size of the media following injury (add data to Fig. 2C).

To address your concern, we have measured the media size in mice with SMC-specific deletion of SRSF1, and confirmed that there is no significant difference between WT and SRSF1-deficient mice. The new data are included in revised **Fig. 2d**.

6.) Fig. 7a+g: neointimal lesions should be co-stained with TUNEL and a specific VSMC marker to confirm that apoptotic cells are indeed VSMC.

Thank you for the suggestion. We have performed co-staining of TUNEL and a specific VSMC α -actin in neointimal lesions, and confirmed that the apoptotic cells are indeed VSMCs (revised **Supplementary Figure 16**).

7.) Since it was reported that delta133p53 promotes angiogenesis it is likely that a knock down or knock out will also affect endothelial cell function during re-endothelialization. This in fact may have a profound impact on neointima formation. In this regard the authors should also assess the

re-endothelialization following vascular injury in this model. Inhibition or knock down/knock out of SRSF1 might result in incomplete re-endothelialization and vessel healing despite a reduced neointimal lesion size. Such a lesion might be more thrombogenic and would prevent such a translational approach from being followed up in further pr-clinical and clinical studies. Thus the effects of SRSF1 on endothelial function and recovery in vitro and in vivo have to be included in the study.

Your points are well taken. We have now investigated the effect of SRSF1 on endothelial cell function. As you predicted, the re-endothelialization of injured arteries from SRSF1-deficient mice was incomplete despite reduced neointimal formation (revised **Figs 2g-h** and **3g-h**). In cultured human coronary artery endothelial cells, Ang II-induced migration and proliferation were promoted by SRSF1 (revised **Fig. 8f-i**), and they were suppressed by SRSF1-knockdown (revised **Fig. 8j-n**). Indeed, this pro-proliferative and pro-migratory effect of SRSF1 on endothelial cells might be an obstacle to developing SRSF1 as a therapeutic target for intimal hyperplasia. Thus, treatment limiting endothelial cell migration and proliferation or anti-thrombotic therapy should be included in potential translational studies targeting SRSF1, as discussed in our revised manuscript.

8,) PCNA staining to determine the number of proliferating cells after injury of the mouse artery were performed at 28 days after injury. However, this is a time point where usually vascular remodelling and VSMC proliferation is completed in this model. Therefore it is quite astonishing that proliferative VSMC can be found. Experiments to determine VSMC proliferation in cross sections of injured mouse arteries should be repeated at 10-14 days after injury.

Per your thoughtful suggestion, we have performed experiments in cross-sections of injured mouse arteries at 14 days after injury to determine VSMC proliferation in SRSF1-deficient mice. Our data

showed that wire injury induced intimal hyperplasia at 14 days post-injury, and the neointima formation was completely suppressed (not detectable) in SRSF1-deficient mice (revised **Fig. 2c,d**), suggesting that SRSF1 deletion inhibits injury-induced VSMC proliferation. We also present the results at 28 days here since neointima was not detectable at 14 days post-injury in SRSF1-deficient mice, and PCNA-positive cells were observed at 28 days in our wire injury mouse model, a similar phenomenon that was reported in recent study by Li group (Zhang et al., 2014).

9.) The manuscript suffers from a general lack of correct grammar and language and should be extensively revised by a native speaker.

We very much appreciate your suggestion. Dr. I.C. Bruce has extensively revised the manuscript.

10.) Migration of VSMC is a further key feature of pathologic vascular remodelling following injury. Thus, the authors should determine the effects of SRSF1-induced $\Delta 133p53$ expression on VSMC migration.

Per your thoughtful suggestion, we have thoroughly investigated the effects of SRSF1-induced $\Delta 133p53$ expression on VSMC migration. Our data showed that overexpression of SRSF1 enhanced VSMC migration (revised **Fig. 4i & Fig. 5j**), and this enhancement was mediated by $\Delta 133p53$ in both human aortic and coronary arterial SMCs (revised **Fig. 5t & Fig. 9m**).

Minor points:

p.1 l.1: correct "KFL5 pathway..." to „KLF5 pathway“

p.2 l.72: "...expression is increased in proliferating VSCM..."

p.5 l.121: “..overexpression was almost double that in the Ad-GFP.. “

p.6 l.143: „In mice, an isoform similar to human $\Delta 133p53$ is highly homologous, it was previously reported...”

p.14 l.327: “...and contributes to the generation of neointima ... “

p.18 l.422: in the methods the authors indicate that “None of the cell lines used were reported in the ICLAC database of commonly misidentified cell lines. All cell lines were tested monthly for mycoplasma contamination and found negative. After their initial purchase, cell lines were not further authenticated. “

However, it is unclear what cell lines are ment since the authors describe that only primary cells were used for all experiments?

We very much appreciate your suggestions. In the revised manuscript, we have deleted unnecessary words and corrected the wording and grammatical mistakes.

References

- Bourdon, J.C. 2007. p53 Family isoforms. *Curr. Pharm. Biotechnol.* 8:332-336.
- Dong, J.T., and C.Chen. 2009. Essential role of KLF5 transcription factor in cell proliferation and differentiation and its implications for human diseases. *Cell Mol. Life Sci.* 66:2691-2706.
- Fujita, K., A.M.Mondal, I.Horikawa, GH.Nguyen, K.Kumamoto, J.J.Sohn, E.D.Bowman, E.A.Mathe, A.J.Schetter, S.R.Pine, H.Ji, B.Vojtesek, J.C.Bourdon, D.P.Lane, and C.C.Harris. 2009. p53 isoforms $\Delta 133p53$ and p53beta are endogenous regulators of replicative cellular senescence. *Nat. Cell Biol.* 11:1135-1142.
- Khoury, M.P., V.Marcel, K.Fernandes, A.Diot, D.P.Lane, and J.C.Bourdon. 2013. Detecting and

quantifying p53 isoforms at mRNA level in cell lines and tissues. *Methods Mol. Biol.* 962:1-14.

Llorian, M., C.Gooding, N.Bellora, M.Hallegger, A.Buckroyd, X.Wang, D.Rajgor, M.Kayikci, J.Feltham, J.Ule, E.Eyras, and C.W.Smith. 2016. The alternative splicing program of differentiated smooth muscle cells involves concerted non-productive splicing of post-transcriptional regulators. *Nucleic Acids Res.* 44:8933-8950.

Midgley, C.A., C.J.Fisher, J.Bartek, B.Vojtesek, D.Lane, and D.M.Barnes. 1992. Analysis of p53 expression in human tumours: an antibody raised against human p53 expressed in Escherichia coli. *J. Cell Sci.* 101 (Pt 1):183-189.

Murray-Zmijewski, F., D.P.Lane, and J.C.Bourdon. 2006. p53/p63/p73 isoforms: an orchestra of isoforms to harmonise cell differentiation and response to stress. *Cell Death. Differ.* 13:962-972.

Roos, G, GLandberg, J.P.Huff, R.Houghten, Y.Takasaki, and E.M.Tan. 1993. Analysis of the epitopes of proliferating cell nuclear antigen recognized by monoclonal antibodies. *Lab Invest* 68:204-210.

Wei, J., J.Noto, E.Zaika, J.Romero-Gallo, P.Correa, W.El-Rifai, R.M.Peek, and A.Zaika. 2012. Pathogenic bacterium Helicobacter pylori alters the expression profile of p53 protein isoforms and p53 response to cellular stresses. *Proc. Natl. Acad. Sci. U. S. A* 109:E2543-E2550.

Yonemitsu, Y., Y.Kaneda, S.Tanaka, Y.Nakashima, K.Komori, K.Sugimachi, and K.Sueishi. 1998. Transfer of wild-type p53 gene effectively inhibits vascular smooth muscle cell proliferation in vitro and in vivo. *Circ. Res.* 82:147-156.

Zhang, S.M., L.H.Zhu, H.Z.Chen, R.Zhang, P.Zhang, D.S.Jiang, L.Gao, S.Tian, L.Wang, Y.Zhang, P.X.Wang, X.F.Zhang, X.D.Zhang, D.P.Liu, and H.Li. 2014. Interferon regulatory factor 9 is critical for neointima formation following vascular injury. *Nat. Commun.* 5:5160.

Reviewers' comments:

Reviewer #1 (Remarks to the Author):

The authors have responded to the critique quite well, having carried out considerable extra experimental work and made appropriate textual alterations.

I am satisfied that the Supplementary Fig 18 addresses my concerns about antibody specificity for SRSF1 and PCNA. I am also satisfied that the various minor issues I raised have been addressed, particularly citing correct papers.

I still have a few concerns about other data:

1. In supplementary Fig 19, panel 4d - siSRSF1 reduces D133p53 as the authors showed in the previous version of the paper, but FLp53 also declines. This begs the question of specificity and needs to be commented on. Similarly, in 4h the siSRSF1 appears to affect FLp53 levels. Being confident that SRSF1 only affects the D133p53 is critical to this paper.

In 4f, in the D133p53 tracks, there is a second band running just below FLp53 - what is this? D40p53? This needs to be commented on.

The figure legend is poor - I had difficulty following the experiments. Finally, the panels should be labelled S19b, c etc; not 4.

2. I also requested that over expression experiments be done with D40p53 and FLp53 to show that the proliferative stimulation of VSMCs is specific to D133p53. In response, the authors say they have done this and it shows the opposite results to D133p53. Great. But I cannot see these data in the revised Figure 4 - panels g-m seem to refer to something completely different and D40 and FLp53 are not mentioned in the Figure legend. I cannot find the result in any of the supplementary data either? Where are the data?

Reviewer #2 (Remarks to the Author):

The authors have added all requested additional experiments and responded sufficiently to most raised questions and concerns. Thus, the authors largely improved the quality of the manuscript. The only remaining concern is that the lacking evaluation of a molecular mechanism responsible for the observed effects still limits the very careful functional and morphological observations.

Point-by-point Reply

Reviewer #1

The authors have responded to the critique quite well, having carried out considerable extra experimental work and made appropriate textual alterations.

I am satisfied that the Supplementary Fig 18 addresses my concerns about antibody specificity for SRSF1 and PCNA. I am also satisfied that the various minor issues I raised have been addressed, particularly citing correct papers.

I still have a few concerns about other data:

1. In supplementary Fig 19, panel 4d - siSRSF1 reduces D133p53 as the authors showed in the previous version of the paper, but FLp53 also declines. This begs the question of specificity and needs to be commented on. Similarly, in 4h the siSRSF1 appears to affect FLp53 levels. Being confident that SRSF1 only affects the D133p53 is critical to this paper.

In 4f, in the D133p53 tracks, there is a second band running just below FLp53 - what is this? D40p53? This needs to be commented on.

The figure legend is poor - I had difficulty following the experiments. Finally, the panels should be labelled S19b, c etc; not 4.

Your points are well taken. We agree with you that whether siSRSF1 affects FLp53 levels is critical in our paper. To fully address your concern, we calculated the FLp53 levels in HASMCs with SRSF1-knockdown by siRNAs as well as in vessels from SRSF1-knockout and WT control mice. Our data showed that the average FLp53 level did not change with SRSF1-knockdown (reply **Figure 1a**) or SRSF1-knockout (reply **Figure 1b**). More representative samples are shown accordingly in revised **Figure 5d,e** and revised **Supplementary Figure 19b,c**.

To identify the ‘*second band below FLp53*’, a band around 45 KD recognized by p53 (CM1) antibody, we overexpressed Ad- Δ 40p53 in parallel with Ad- Δ 133p53 in HASMCs. Western blots using p53 (CM1) antibody showed the ‘*second band*’ had the same molecular weight as Δ 40p53 (reply **Figure 2a**). Since p53(CM1) is a rabbit polyclonal antibody that recognizes all p53 isoforms (Bourdon, Jean-Christophe. *p53 and its isoforms in cancer. British Journal of Cancer, 97.3 (2007):277-282*), and isoforms p53 β and p53 γ share molecular weights close to Δ 40p53, we used another antibody, DO-1, to detect the ‘*second band*’. The mouse monoclonal p53(DO-1) antibody is specific to the N-terminal domain (amino-acids 20-25) in human p53 which is not present in Δ 40p53 and Δ 133p53, so it only recognizes p53, p53 β , and p53 γ (Bourdon, Jean-Christophe. *p53 and its isoforms in cancer. British Journal of Cancer, 97.3 (2007):277-282*). Our data showed that the ‘*second band*’ differently expressed in the Ad-GFP and Ad- Δ 133p53 tracks was not revealed by p53 (DO-1) antibody detection (reply **Figure 2b**), excluding the possibility of it being p53 β or p53 γ . Taken together, from the aspects of molecular weight and its detectability by the p53 (CM1) but not the p53 (DO-1) mAb, this Ad- Δ 133p53-induced ‘*second band*’ might be Δ 40p53.

Per your suggestion, the panels in Supplementary Figure 19 have been labelled Supplementary Figure 19a-e, and the legend has been rewritten accordingly (revised Supplementary Fig. 19).

2. I also requested that over expression experiments be done with D40p53 and FLp53 to show that the proliferative stimulation of VSMCs is specific to D133p53. In response, the authors say they have done this and it shows the opposite results to D133p53. Great. But I cannot see these data in the revised Figure 4 - panels g-m seem to refer to something completely different and D40 and FLp53 are not mentioned in the Figure legend. I cannot find the result in any of the supplementary data either? Where are the data?

We really apologize for the confusion. All the data on $\Delta 40p53$ are presented in revised Figure 5g-m.

REVIEWERS' COMMENTS:

Reviewer #1 (Remarks to the Author):

I believe the authors have now satisfied all my concerns.